# Over a century of global decline in the growth performance of marine fishes

Helen F. Yan [1,2] ✉, Hannah V. Watkins [3], Alexandre C. Siqueira [1,4] & David R. Bellwood [1]

Human-driven pressures are causing large-scale changes in the ecologies and life histories of fishes. Growth performance is a composite life-history trait that captures the trade-off between two fundamental traits: growth and body size. Here, we assess the impacts of fishing and temperature on the growth performance of marine teleost fishes globally over the last century. Using 7683 growth curves encompassing 1479 species, we find a global pattern of decline in growth performance from 1908 onwards, with the greatest declines in commercially valuable fishes. Indeed, managed fisheries experienced a 9% decline in growth performance over the last century, which can equate to an average decline of up to 23% in asymptotic body size or a 45% decline in the von Bertalanffy growth coefficient ($K$). Despite relatively consistent increases in ocean temperatures globally, we only detect a decline in the growth performance of fishes in temperate regions, which is probably indicative of an overrepresentation of commercially valuable fishes at higher latitudes. The declines in growth performance likely reflect legacy effects, shaped by overfishing, on changes in the underlying size structure and demographic processes of fished stocks. Therefore, the potential impacts of warmer temperatures on growth performance may be overwhelmingly masked by the impacts of overfishing.

We are currently in an era of global change. The oceans of the Anthropocene are becoming increasingly impacted by destructive, human-driven activities, threatening critical ecosystem functions and services. Undoubtedly, the two greatest threats to ocean biodiversity are overexploitation and biophysical changes due to climate change[1–3]. The extinction risk of many of the world's most iconic and culturally valuable fishes is increasingly elevated by overfishing[4–7]. Although fisheries management regimes can improve the status of fished stocks when implementation is strong[8–10], many of the world's managed fisheries are still overfished[11,12]. This paradigm is likely an artifact of the temporal mismatch between overfishing and management: strong management is typically triggered once stocks have been seriously depleted[13–15]. Consequently, large declines in stock abundances have not only led to declines in fisheries production but have also led to demographic changes in the life histories of many commercially valuable species[16–20].

Changes in fishes' life histories may be further exacerbated by changes in environmental conditions due to climate change. The impacts of climate change are manifesting throughout the world's oceans, with increasing temperatures and decreasing dissolved oxygen levels[21] driving changes in relatively plastic critical life-history traits. There is substantial evidence that the expression of life histories differs between conspecifics found across different temperature regimes, with faster life-history traits (e.g. faster growth rates, earlier

[1]Research Hub for Coral Reef Ecosystem Functions, College of Science and Engineering, James Cook University, Townsville, QLD, Australia. [2]Thriving Oceans Research Hub, School of Geosciences, University of Sydney, Camperdown, NSW, Australia. [3]School of Resource and Environmental Management, Simon Fraser University, Burnaby, BC, Canada. [4]Centre for Marine Ecosystems Research, School of Science, Edith Cowan University, Perth, WA, Australia. ✉e-mail: helen.yan@sydney.edu.au

ages at maturity, smaller body sizes) typically coinciding with warmer temperatures[22–24]. Climate change can thus push the physiological limits of fishes towards decreases in body size[25] or range expansions into deeper waters and/or higher latitudes[25–27]. The potentially compounding synergism between fishing and climate change may disproportionally impact the population dynamics of the world's fishes more than any one stressor alone, driving changes in their ecologies and life histories[2]. The impacts of climate change and over-fishing have now extended to all reaches of the ocean[3], yet we are lacking a global evaluation of their respective impacts on the life histories of fishes.

To address these issues, we use a global dataset (Fig. 1a) comprising 7683 growth curves of marine teleost fishes, which encompasses 1479 species over 113 years[28]. We then implement a suite of Bayesian state-space models, which separate the underlying population state from observation deviances[29], to reveal global trends in growth performance over the last century (from 1908 to 2021). Growth performance ($\phi$; sensu ref. 30) is a composite life-history trait[31] that

captures the trade-offs between asymptotic body size and the von Bertalanffy growth coefficient ($K$), two dominant and fundamentally linked life-history traits[32,33] (Fig. S1). Specifically, growth performance standardises estimates of instantaneous growth coefficients against a single unit of body size[30]. Using the temporal trends of growth performance, we can then determine if changes coincide with increasing temperatures and/or different fisheries management regimes. First, we assess the global growth performance of fishes from 1908 to 2021. Second, we adapt a management classification scheme[34] to disaggregate these patterns by managed fisheries, unmanaged fisheries, and unfished species. Managed fisheries represent families for which formal stock assessments are available, whereas unmanaged fisheries encompass families that have global catch landings reported or reconstructed but lack formal stock assessments; the remaining families are classified as "unfished". We then assess the spatial patterns in growth performance by separating the time series into temperate, subtropical, and tropical locations, to account for potentially different geographic patterns. Finally, we directly assess the combined impacts

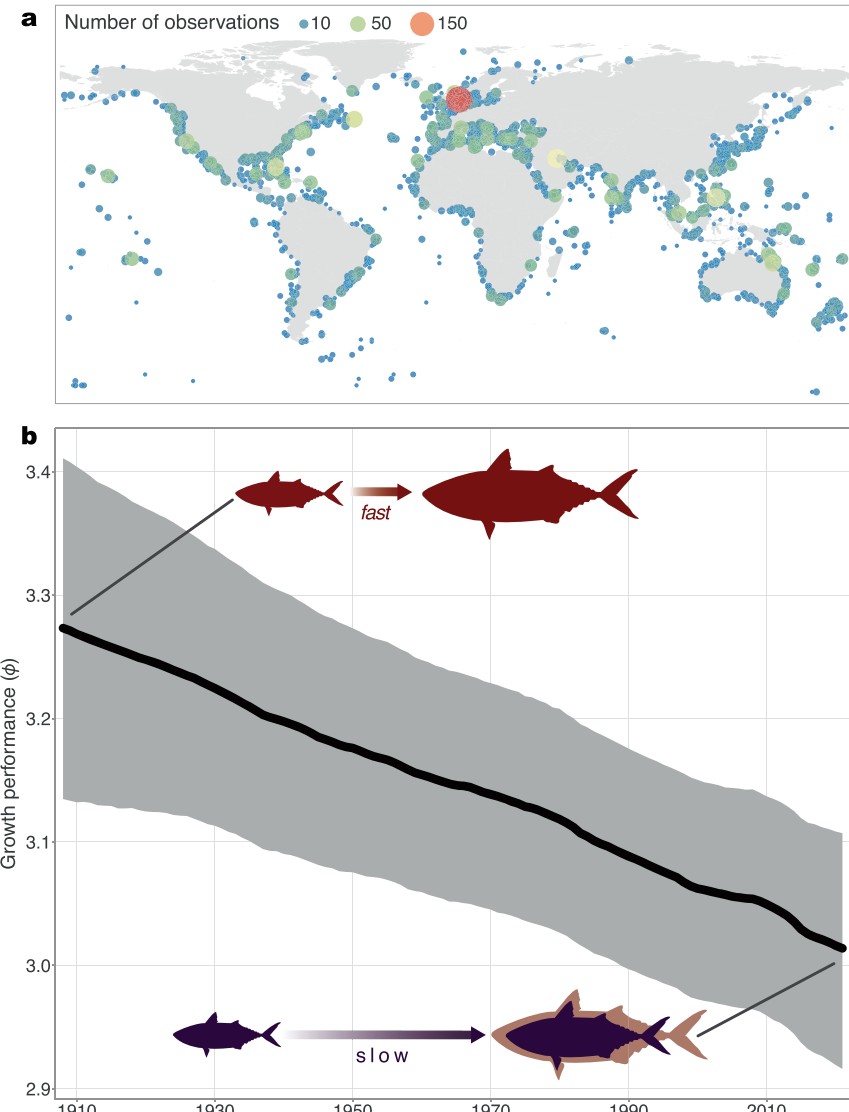

**Fig. 1 | Distribution of 7683 growth curves indicate a global decline in fish growth performance. a** The distribution of all collated growth curves. Points indicate individual observations and are coloured and sized (area) based on the number of observations per location. **b** The global growth performance ($\phi$) of marine fishes from 1908 to 2021 ($n = 7683$ observations). The grey ribbon denotes

the 90% credible interval and the thick black line is the median modelled estimate. Growth performance values around 1908 represent species that grow relatively quickly to larger body sizes, whereas lower growth performance values denote species that grow relatively slower to smaller body sizes. Source data are provided as a Source data file.

of fishing and temperature on growth performance, after accounting for phylogenetic relatedness.

## Results and discussion
### Global decline of growth performance
Globally, we found that the average growth performance across 1479 focal species had declined from 1908 to 2021 by 7.9% (90% credible interval: 4.4% to 11.4%; Fig. 1b). The median predicted growth performance in 1908 was 3.3 (3.1–3.4) compared to 3.0 (2.9–3.1) in 2021, which could be indicative of declines in $K$, $L_\infty$, or both (see Methods). For instance, if $L_\infty$ remained unchanged, this would indicate a 44.9% (29.0–60.0%) decline in $K$; if $K$ remained unchanged, this decline in growth performance would represent a 22.8% (13.2–32.2%) decline in $L_\infty$. Notably, in both cases (i.e. a change in $L_\infty$ or a change in $K$), the size for a given age is reduced (Fig. S2). This global decline is most likely driven by fishes that are growing to relatively smaller sizes and/or at slower rates (i.e. intraspecific changes), and less likely driven by compositional shifts in the species sampled (i.e. interspecific changes; Fig. S3). Given the spatial disparities in fishing pressure and ocean warming around the world, we can further disaggregate these global patterns across different management regimes.

### Overfishing is driving global declines
When we disaggregated the global time series by management regime, we found that fishes from managed fisheries exhibited the greatest declines in growth performance over the last century. We found a steady decline in growth performance of managed fishes from 1908 onwards, declining by 9.1% (5.0–13.2%; Fig. 2a). This resulted in a decline of a median fitted growth performance of 3.5 (3.4–3.7) in 1908 to 3.2 (3.1–3.3) in 2021, corresponding to an average asymptotic body size decline of 27.2% (16.2–38.8%) when holding $K$ constant or a 51.9% (34.5–68.8%) decrease in the growth coefficient $K$ when $L_\infty$ is constant over the last century. Unmanaged and unfished species exhibited no change in their respective growth performances through time (Fig. 2b, c). We found that the growth performance of unmanaged fishes, although lower than most managed fishes at a mean of 3.0, exhibited no net change across the time series (−11.0–21.6%; Fig. 2b). Similarly, the growth performance of unfished species remained relatively stable as well (−20.7–44.6%; Fig. 2c).

The decline in growth performance in managed fisheries is not indicative of the role of management per se; it is likely a reflection of the legacy of historically intense fishing mortality experienced by commercially valuable fishes. Indeed, declines in the fished biomass of commercially valuable fisheries were similarly matched with increases in fishing effort until the mid-1990s[10], which can be an indication of overfishing. Similarly, the total reported and reconstructed catches of managed fisheries had increased steadily until the mid-1980s before plateauing[35], whereas the number of stock assessments only started to increase after the 1990s[36]. Although the catches of unmanaged fishes followed a similar pattern to managed fishes, the catch of unmanaged fisheries is approximately half that of managed fisheries[35] (Fig. S4). Many commercially valuable fishes had been heavily exploited well before management began, indicating that management is often a reaction to decline as opposed to a preventative measure[37]. The presence of management alone is not an indication of effective management; some of the most heavily regulated fisheries in the world still resulted in significant overfishing and population collapse (e.g. refs. 38,39.). Additionally, some stocks that are effectively managed may not be overfished but may experience prolonged fishing pressures (i.e. maximally fished), which can similarly lead to fishing-induced demographic changes[20]. Although effective fisheries management practices can improve stocks that are in decline[8–10], it may take longer to achieve similar levels of recovery in the life histories of historically exploited fishes.

The declining growth performance of commercially valuable fishes is likely a legacy effect of size-based targeting via fisheries. Fishers are incredibly effective hunters and tend to target the largest (and typically oldest) individuals of a stock[40]. Unsurprisingly, intense fishing pressure has truncated populations towards smaller average sizes and younger ages[20,41]. The resulting populations characterised by younger individuals (i.e. population juvenescence) not only express faster life histories, such as smaller body sizes and/or faster growth rates, but also exhibit less stability through time[20]. The dynamic instability of juvenescent populations is likely driven by changes in demographic compensatory mechanisms, such as increased somatic growth or fecundity[20], or potentially through the loss of social learning[42]. An alternative but not mutually exclusive mechanism for the observed declines in growth performance could be due to the

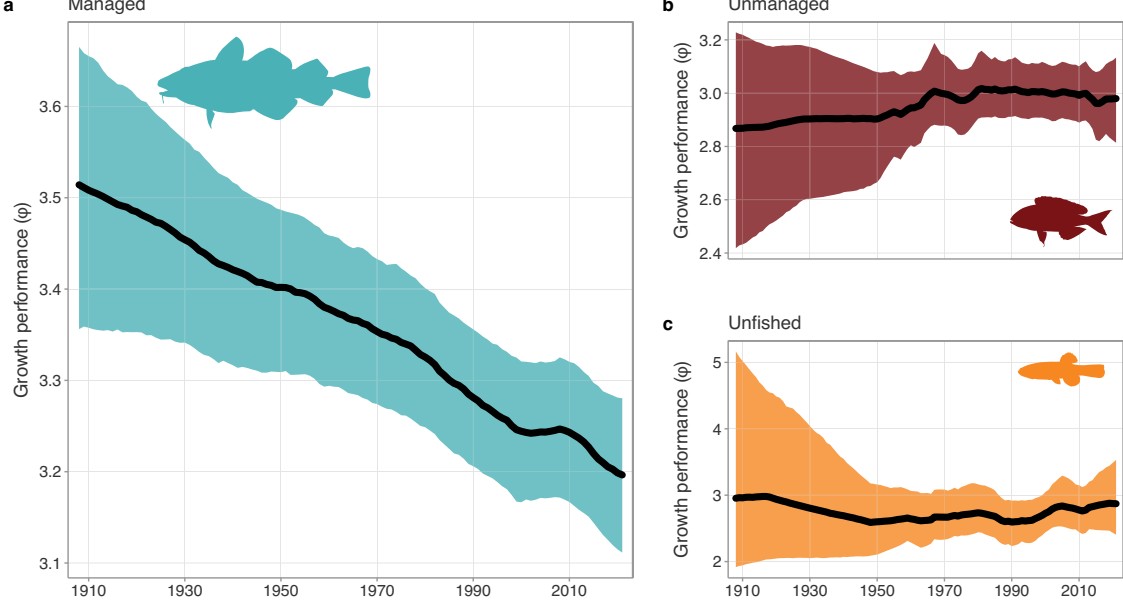

**Fig. 2 | Changes in growth performance through time by management status.** Changes in growth performance ($\phi$) of fishes in **a** managed fisheries ($n = 996$ species), **b** unmanaged fisheries ($n = 419$), and **c** unfished species ($n = 64$) from 1908 to 2021. The thick ribbons denote the 90% credible intervals, and the thick black lines are the median fitted trends. Source data are provided as a Source data file.

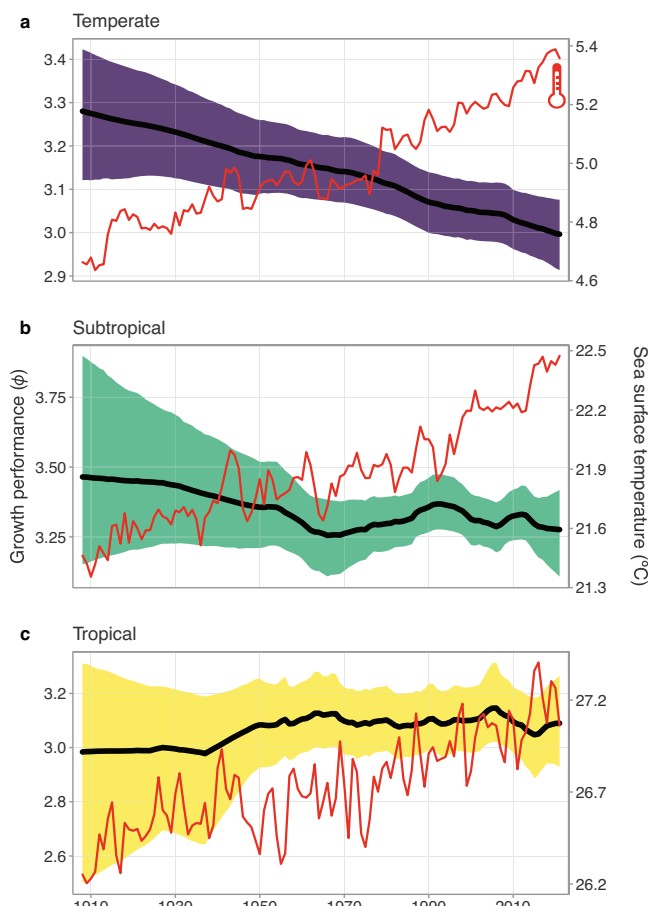

**Fig. 3 | Changes in growth performance and increasing ocean temperatures.**
Changes in growth performance ($\phi$) of fishes in **a** temperate ($n = 3536$ observations), **b** subtropical ($n = 1483$), and **c** tropical regions ($n = 2664$) from 1908 to 2021 (left $y$ axis). The thick ribbons denote the 90% credible intervals of the time series and the thick black lines are the median estimates. The red line in each panel corresponds to the mean sea surface temperature (°C) in each respective region for the same time period (right $y$ axis). Source data are provided as a Source data file.

'Rosa Lee' phenomenon[43]. In short, larger, faster-growing individuals are targeted first by size-selective fisheries, allowing the smaller, slower-growing individuals of a population to survive[44]. Consequently, for a given age, the remaining individuals are smaller and exhibit slower growth rates than would be expected in the absence of size-selective fishing[44], potentially resulting in declines in growth performance. Therefore, the declines in growth performance of commercially valuable fishes over time could represent the interplay between two synergistic processes. First, an unfinished, residual stock is remaining with smaller individuals from size-selective fishing, which may be dominated by slow-growing individuals (via the 'Rosa Lee' phenomenon[43]). Second, these demographic changes are then further reinforced and amplified by fisheries-induced selection against larger body sizes, which would manifest as lower growth performance values in subsequent generations[45–48].

Declines in global growth performance can be indicative of both intraspecific changes and interspecific compositional changes. Despite considerable overlap in the families and species that were sampled through time, which suggests relative homogeneity in the annual species composition, there were also relatively distinct compositional shifts between specific time periods (Fig. S3). While it is plausible that declines in growth performance could be due to potential sampling biases (i.e. shifting from species with higher growth performance values towards those with lower growth performance through time), it

is unlikely that this is the primary mechanism of the observed declines. Indeed, there is considerable overlap between the early (pre-1960) and late (post-2000) compositional assemblages, which have the most leverage on temporal trend estimates (Fig. S3). Additionally, species compositions explained very little variation across growth performance studies (Fig. S3). It is therefore unlikely that the observed declines in growth performance were solely due to changes in the species sampled through time. Irrespective of the underlying mechanism, given the inherent temporal lag between management implementation and stock recovery[12], the legacy of size-based fishing could manifest throughout populations as declines in growth performance, even when abundances are stable or increasing. Regardless of whether overfishing is the proximate or distal cause for the decline in fish growth performance, changes in the life histories of fishes are likely to be slow to reverse (or potentially irreversible) and can have repercussions on the biomass and productivity of capture fisheries[45,47].

### Geographic patterns reflect overfishing

Disaggregating the time series by geographic location largely reflected patterns in fisheries management. Growth performance in temperate locations declined by 8.7% (90% credible interval: 4.2– 13.2%; Fig. 3a). Although the decline in growth performance does coincide with an increase in average ocean temperature of 0.7 °C over the same time period (from 4.7 °C in 1908 to 5.4 °C in 2021), we found generally no change in growth performance in subtropical regions (−17.5–4.9%), which experienced an even greater level of warming (from 21.5 °C in 1908 to 22.5 °C in 2021; Fig. 3b). Similarly, despite stronger temperature oscillations in the tropics, we also found no change in the growth performance of tropical fishes (−10.3–20.8%; Fig. 3c). The decline in growth performance of fishes in the temperate region is likely a manifestation of the dominance of managed fisheries in higher latitudes[10], particularly in the northern hemisphere (Figs. 4 and S5). Indeed, nearly 90% of the studies in the temperate region comprises managed fisheries (Fig. S6) and the strongest evidence of decline was found when separate analyses were run for managed fisheries in temperate and subtropical regions (Fig. S7). The dominance of commercially valuable, and thus managed, fisheries is likely the primary factor driving the decline in the growth performance of fishes in the temperate region, which appear to have extended to some subtropical fishes as well (Fig. S7).

The impacts of commercial size-based fishing are likely confounding and masking the potential physiological impacts of rising temperatures on the growth performance of fishes. After accounting for phylogenetic relatedness, we found strong evidence of a positive relationship between temperature and growth performance in both unmanaged and unfished species (median estimate [90% credible interval]: 0.023 [0.0099 to 0.036] and 0.040 [−0.00076–0.081], respectively). For unmanaged fishes, a 1 °C increase in temperature translated to, at most, a 7–15% median increase in $L_\infty$ or an 18–37% increase in $K$ across the majority of the range of observed temperatures (smaller estimates calculated from 0 to 1 °C and larger estimates from 28 to 29 °C). For unfished species, a similar increase in temperature resulted in a maximum increase of 14–48% in $L_\infty$ or a 34–147% increase in $K$. Across managed fishes, however, we found no evidence of an effect of temperature on growth performance (−0.00011 [−0.0045–0.0050]; Fig. 5). It appears that larger temperature gradients can be a strong driver of growth across fishes; however, a single degree of warming may be too small to lead to detectable changes in growth performance over an entire century. Indeed, temperature has little visible impact on commercially valuable, managed fishes, further reinforcing size-selective fisheries as a dominant driver of declines in growth performance.

Across all three regions (i.e. temperate, subtropics, and tropics), the temperature has increased steadily by roughly 1 °C across the entire time series. Increases in ocean temperatures, paired with

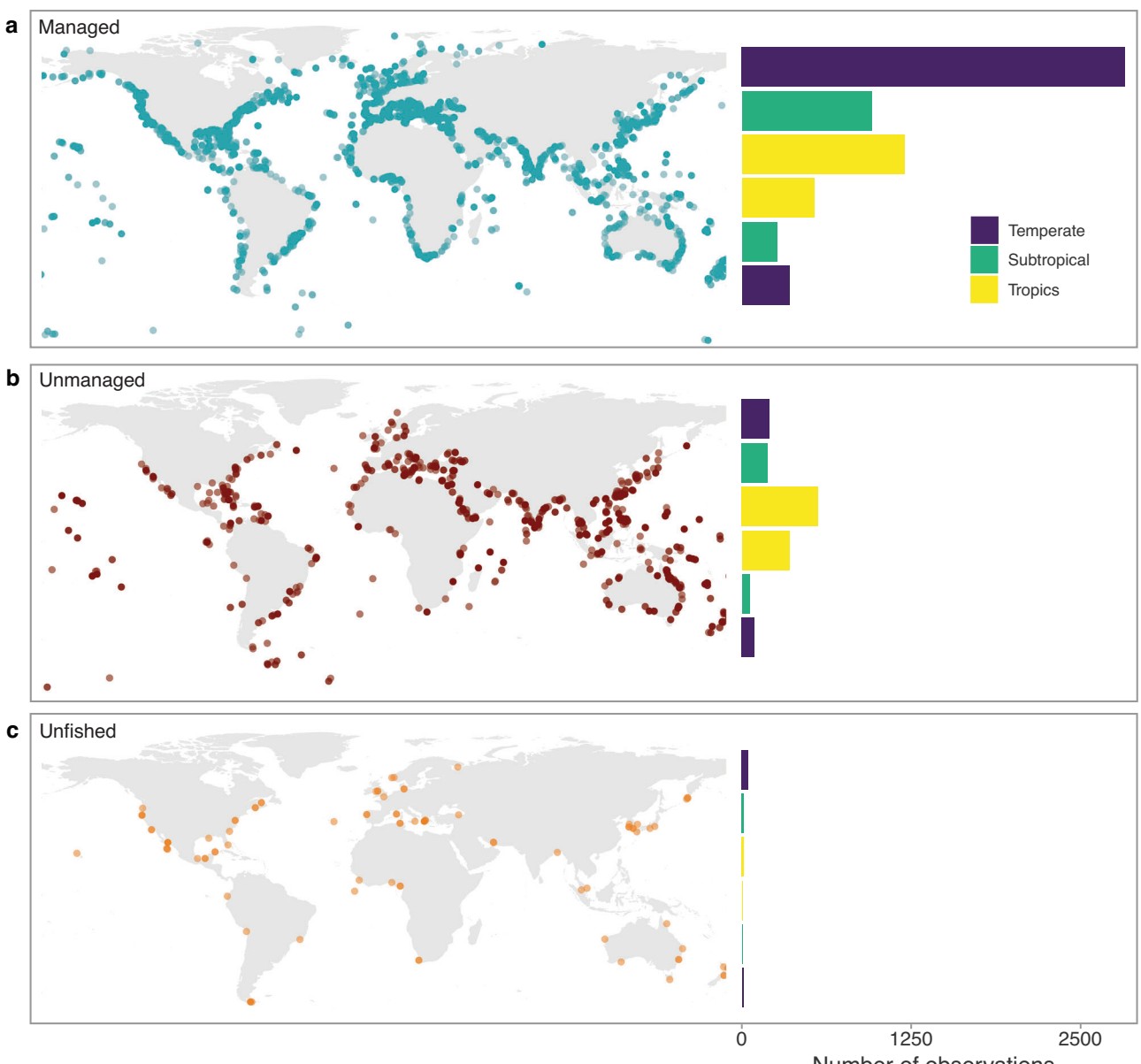

**Fig. 4 | Global distribution of fisheries management.** The spatial distribution (left) and total number of growth performance studies (right) from **a** managed fisheries (top; green), **b** unmanaged fisheries (middle; red), and **c** unfished species (bottom; orange). Total number of observations per region are separated by the northern and southern hemisphere. Source data are provided as a Source data file.

declines in dissolved oxygen[21], can push the physiological limits of ectotherms, such as fishes, towards faster metabolic rates[25,49,50]. Given the inherent correlation between metabolic rates and growth[31,51], models of energy assimilation predict that under a high-emissions climate scenario, the asymptotic body sizes of fishes may decrease by 14–24% globally in the next 30 years due to changes in both physiology and species assemblages[25]. Empirically, while not directly comparable, species' average body sizes have been shown to both decrease and increase with warming temperatures[52], indicating that changes in body size can occur relatively quickly (over only a few decades), but in potentially polarising directions. Nevertheless, fishing and warmer temperatures may act synergistically to drive further changes in fishes' body sizes. Declines in body size across commercially valuable fishes would not only have demographic consequences with potential flow-on effects on the functioning of marine systems, but would also impact fisheries yields and global protein supplies[25,37].

Unmanaged fisheries are typically exposed to unsustainable fishing pressures, therefore we might expect that growth performance would follow, if not surpass, the same magnitude of decline found in commercially valuable fishes. However, we found that the growth performance of unmanaged fishes was relatively stable throughout the time series. Although an underwhelming pattern of stagnation could be indicative of resilient demographic processes (i.e. resilience to fishing and temperature changes) or lagged responses, it is also possible that the focus on unmanaged fishes is disproportionately higher in tropical regions and may have biased the current literature on growth across marine fishes. The tropics are dominated by small-scale coral reef fisheries, where management is typically regulated by local communities rather than national-level policies[53]. Consequently, many coral reef nations lack the resources or funding to conduct stock assessments[54] and such unmanaged stocks are thus misrepresented in global fisheries census data[10]. Given that growth studies can be tightly

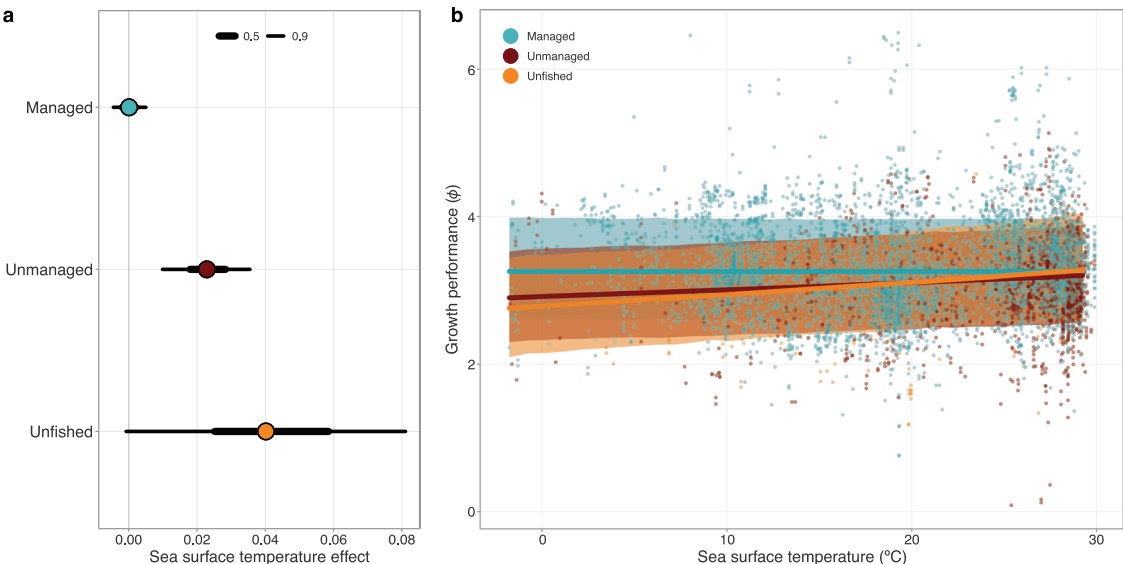

**Fig. 5 | Effects of temperature across managed, unmanaged, and unfished species.** The modelled **a** slope estimates and **b** fitted effects of sea surface temperature on the growth performance of managed (green), unmanaged (red), and unfished species (orange) after accounting for phylogenetic relatedness ($n = 7683$ observations). **a** Points denote the median and thick and thin bars denote the 50% and 90% credible intervals, respectively. Note the 50% credible interval is too narrow to be seen for managed fishes. The ribbons and thick lines in **b** denote the 90% credible interval and the median modelled fits, respectively. The points represent the raw data. Source data are provided as a Source data file.

linked to conventional fisheries management practices[13], the lack of stock assessments in the tropics may have led to an under-representation of unmanaged fishes in the growth literature. Indeed, unmanaged fishes may already be operating on a shifted baseline, whereby stocks are severely depleted to the point of mainly comprising individuals with lower growth performance. Care is needed, therefore, as these systematic biases could distort or completely omit any realised changes in growth performance across unmanaged fishes.

We uncovered global patterns of decline in the growth performance of marine fishes over the last century. The decline in growth performance of temperate fishes is most likely the by-product of an overrepresentation of commercially valuable, managed fishes in this region. Specifically, in the temperate region, managed fishes experienced the strongest magnitude of decline compared to unmanaged and unfished species, which exhibited relative stability through time (Fig. S7). Although we found evidence of an effect of temperature on the growth performance of unmanaged and unfished species, the effect of one degree was not sufficiently strong to produce detectable changes over an entire century. We therefore found stronger evidence that commercial size-based fishing practices, not temperature, are the primary drivers of the global pattern of declining growth performance in marine fishes. The pervasiveness of size-selective fisheries can both change the demographic dynamics of fish populations[20] and trigger relatively strong selection pressures towards smaller body sizes[45–48], both of which could manifest as declines in growth performance across ocean basins. Adaptive fisheries management protocols, such as the holistic implementation of both minimum and maximum size restrictions (i.e. harvest slots)[55], could provide localised levers to managers to prevent further declines in fishes' life histories. In turn, these localised mechanisms could potentially offset the global-scale negative impacts of climate change on the life histories of marine fishes.

## Methods
### Data collation and growth performance
To calculate growth performance, we used the von Bertalanffy growth parameters $L_\infty$ (cm), which is the expected asymptotic size for an individual in a given population, and $K$ (year$^{-1}$), which is the

instantaneous rate at which an individual approaches its asymptotic size[56]. We used the growth data generated in ref. 28, but excluded hagfishes (family Myxinidae) to retain only ray-finned fishes in our analyses. In short, we first generated a list of growth studies from FishBase[57] and manually corrected all detected typological/rounding errors by individually assessing each reference; observations were removed when growth parameters could not be found. We then conducted a systematic literature search using the ISI Web of Science database to find all published articles using von Bertalanffy growth parameters for marine fishes and cross-checked the list with that from FishBase to remove any duplicates (conducted 21 May 2022). We used the search terms 'fish* AND (growth OR von Bertalanffy)', which yielded 141649 articles. Articles were rejected based on the titles and abstracts. We excluded all artificially reared individuals (including aquaculture species), as well as all freshwater species and euryhaline species collected in freshwater habitats. For each study, we recorded the growth parameters $L_\infty$ and $K$, the length measurement type (i.e. total length, standard length, or fork length), the method used for aging (i.e. mark recapture, otolith rings, other growth rings [e.g. scale rings, vertebrae rings], length frequency, or unknown), the year for which individuals were surveyed, and the geographic location for which individuals were collected. Articles missing any of the above-mentioned information were excluded. Following ref. 58, we only included studies where the discrete geographic locations were provided. For example, studies conducted in Chesapeake Bay were included, but those conducted across the entire Atlantic were excluded. In total, we retained 2886 articles, resulting in 7683 growth curves encompassing 1479 species over 113 years (from 1908 to 2021). To make studies comparable, we used published length-length conversion coefficients from FishBase[57] to convert estimates of $L_\infty$ recorded as standard or fork length to total length in cm using the following equation:

$$\text{Total length} = a + b \times \text{standard or fork length} \tag{1}$$

Here, a and b are the length-length conversion factors from FishBase[57]. When species-level coefficients were not available, we used genus or family means.

Using estimates of $L_\infty$ and $K$, we are able to calculate growth performance as defined by ref. 30. Growth performance ($\phi$) is a composite life-history trait (i.e. a life-history trait calculated from other life-history traits)[31] and encompasses the trade-off between the von Bertalanffy growth coefficient and body size[30]. Specifically, it standardises for body size and allows the comparison of growth coefficients across species and/or populations[30,58]. For example, species that have higher growth performance values tend to exhibit higher resting metabolic rates for a given body size[31]. Growth parameters were converted to estimates of growth performance using the equation:

$$\phi = \log_{10} K - S_L \times \log_{10} L_\infty \tag{2}$$

Here, $K$ and $L_\infty$ are the growth parameters and $S_L$ is the slope between $\log_{10} L_\infty$ and $\log_{10} K$ for the population. The parameter $S_L$ can also be defined as

$$S_L = -m \times b \tag{3}$$

Where $m$ is the anabolic term exponent from the von Bertalanffy growth model and $b$ is the length-weight regression exponent. Following refs. 58,59, we used the metabolic scaling exponent in place of the anabolic exponent $m$, which was found to be, on average, 0.761 for fishes[59], for our calculation of $S_L$.

To test the sensitivity of $\phi$ to different $S_L$ values, we re-ran the global time series model but calculated $\phi$ using empirically derived $S_L$ values. We calculated $S_L$ by regressing $\log_{10} K$ against $\log_{10} L_\infty$ for all species-level observations that had more than three observations per aging method (adapted from refs. 33,58). This resulted in 714 species with empirically measured $S_L$ values. We used the 'constant' $S_L$ value for all species without empirically measured estimates by multiplying the mean anabolic term by the length-weight regression exponent (as denoted by Eq. 3). While we found a relatively positive association between both methods of calculating growth performance (i.e. constant vs. varying $S_L$ values; Fig. S8), using varying $S_L$ values produced much more extreme values of growth performance compared to using constant values (−15.43–41.07 vs. 0.087–6.5, respectively), which exceeded the estimates of previous interspecific growth studies[58,60]. Despite these extreme values, we found a steeper, albeit more uncertain pattern of decline in the global time series (median decline [90% credible interval]: 21.0% [−39.3–1.53%]; Fig. S9). We therefore present the conservative estimates generated using the 'constant' $S_L$ values in the main text. See supplementary methods for model specifications.

We used Eq. (2) to generate hypothetical scenarios depicting how changes in growth performance ($\Delta\phi$) across time steps ($t$) would result in changes in either $L_\infty$ and $K$ (while holding $K$ and $L_\infty$ constant, respectively; Fig. S2). Here, we used the "constant" $S_L$ value (see above) with the mean length-weight regression exponent across species ($b = 3.03$).

$$\Delta\phi = \phi_t - \phi_{t-1} = \left[\log_{10} K_t - S_L \times \log_{10} L_{\infty,t}\right] \\ - \left[\log_{10} K_{t-1} - S_L \times \log_{10} L_{\infty,t-1}\right] \tag{4}$$

$$\Delta\phi = \log_{10}\left(\frac{K_t}{K_{t-1}}\right) - S_L \times \log_{10}\left(\frac{L_{\infty,t}}{L_{\infty,t-1}}\right) \tag{5}$$

Solving Eq. (5) to find changes in $K$ while keeping $L_\infty$ constant removes the entire $L_\infty$ term by setting $S_L \times \log_{10}(1) = 0$.

$$\left(\frac{K_t}{K_{t-1}}\right) = 10^{\Delta\phi} \tag{6}$$

Conversely, solving Eq. (5) to find changes in $L_\infty$ produces:

$$\left(\frac{L_{\infty,t}}{L_{\infty,t-1}}\right) = 10^{\frac{-\Delta\phi}{S_L}} \tag{7}$$

## Spatial and management regimes

Using the geographic locations of each growth curve, we were able to (1) separate the locations into different regions and (2) extract estimates of sea surface temperature from each location. We used the latitude of each location to assign locations into different regions (i.e. temperate, subtropical, and tropical). Tropical locations were chosen between −23° and 23°, subtropical were from 23° and 34° in the northern hemisphere and −23° and −34° in the southern hemisphere, and the remaining locations were classified as temperate and included polar regions. Across all three regions, the number of observations noticeably increased from the 1950s to the early 2000s, before steadily decreasing after 2010 (Fig. S10). Sea surface temperature raster files were calculated from annual temperature means at a $1° \times 1°$ resolution from https://psl.noaa.gov/data/gridded/data.cobe.html. Using mean annual temperature raster files and the geographic location of each observation, we created a 2° buffer around each point and calculated mean sea surface temperature per location per sampling year.

To reflect different fishing pressures across our samples, we adapted the method by ref. 34 and separated families into managed fisheries, unmanaged fisheries, and unfished species. Similarly to ref. 34, all families for which a formal stock assessment existed from the RAM Legacy Database[36] was considered a 'managed fishery'[61]; conversely, families that had reported or reconstructed catches on the Sea Around Us database[35], but no formal stock assessment, were then considered 'unmanaged fisheries'. All remaining populations were considered unfished species. Because we made separations at the family level, we acknowledge the potential biases that not all species belonging to a family will be formally managed; however, this is likely to make our results more conservative because species belonging to the same families typically exhibit similar ecologies and are likely similarly vulnerable to fisheries.

## Time series analyses

We developed separate Bayesian state-space models to assess the growth performance of fishes from 1908 to 2021 globally, under different management regimes (i.e. managed, unmanaged, and unfished), and different regions (i.e. temperate, subtropics, and tropics), after accounting for environmental temperature. To assess the changes in growth performance through time, we used a state-space modelling approach treating each growth performance value as an individual observation (taken from a population/stock's calculated growth parameters). Each model estimates the trend in a single underlying population state by accounting for observation error[29]. The growth performance of fishes ($y_i$) in year $t$ was modelled according to a gamma error distribution by specifying the gamma shape ($\tau$) and rate parameters ($\lambda_{i,t}$).

$$\text{Gamma}(y_{i,t}|\tau, \lambda_{i,t}) \tag{8}$$

$$\lambda_{i,t} = \frac{\tau}{\mu_{i,t}} \tag{9}$$

$$\log(\mu_{i,t}) = \alpha_t + \alpha_{fam} + \beta X_i \tag{10}$$

Here, the term $\alpha_{fam}$ is the varying effect of family on the growth performance of fishes. The term $\beta$ is the estimated population-level effect of temperature and $X$ is the extracted temperature for each specified location for each growth performance value $y_i$. We set the

following weakly informative priors:

$$\lambda \sim \text{Gamma}(0.01, 0.01) \tag{11}$$

$$\beta \sim \text{Normal}(0, 2) \tag{12}$$

The term $\alpha_t$ ($t = 1908, \ldots 2021$) is the expected mean growth performance for each year. Each annual growth performance estimate is considered the observed estimate of an underlying, unobserved population state $x_t$. The underlying population state $x_t$ changes from year-to-year following a first-order autoregressive process, where

$$x_t = x_{t-1} + u + w_t \tag{13}$$

$$\alpha_t = x_t + v_t \tag{14}$$

$$w_t \sim \text{Normal}\left(0, \sigma_q\right) \tag{15}$$

$$v_t \sim \text{Normal}(0, \sigma_r) \tag{16}$$

Here, $w_t$ represents the random annual deviations (i.e. process deviance) in growth performance resulting from demographic processes of the population and is drawn from a normal distribution with a mean of zero and a process standard deviation of $\sigma_q$ (variance). The drift term $u$ allows the model to estimate if there is a consistent trend upwards or downwards over time. The 'observed' growth performance of a given year represents the underlying state $x_t$ plus some observation error $v_t$ (i.e. observation deviance), which is drawn from a normal distribution with a mean of zero and its respective measurement standard deviation $\sigma_r$. To estimate the state at $x_{t=1908}$, we estimated the initial state $x_0$ following the same equation as above.

$$x_{t=1} = x_0 + u + w_{t=1} \tag{17}$$

The initial state $x_0$ and the drift term $u$ used the following weakly informative priors:

$$x_0 \sim \text{Normal}(0, 5) \tag{18}$$

$$u \sim \text{Normal}(0, 5) \tag{19}$$

Individual aging methods (i.e. mark recapture, otolith rings, other rings [e.g. scales, vertebrae], length frequency analyses, and unknown) each contain their own mechanistic biases, which can lead to different values of $L_\infty$ and $K$ (and thus, different growth performance values $\phi$)[58,62]. Indeed, aging method accounts for roughly 5% of the variation in growth coefficients across fishes[28]. To account for these inherent methodological biases quantitatively, each method was modelled with its own respective observation deviance. Annual estimates of expected growth performance of the same underlying population state $x_t$ may have constant differences between different aging methods. Therefore, we added scalar terms for four of the aging methods (i.e. mark recapture ($c_{MR}$), other rings ($c_{OR}$), length frequency ($c_{LF}$), and unknown ($c_{UN}$)) to scale estimates towards a baseline method (i.e. otoliths). All scalar terms were specified with weakly informative normal priors:

$$c_{MR}, c_{OR}, c_{LF}, c_{UN} \sim \text{Normal}(0, 5) \tag{20}$$

Each aging method also included its own respective measurement error in its estimate of the same underlying population state $x_t$ (i.e. $v_{OT}$, $v_{MR}$, $v_{OR}$, $v_{LF}$, $v_{UN}$; OT = otolith rings, MR = mark recapture, OR = other rings, LF = length frequency, and UN = unknown).

$$\begin{bmatrix} \alpha_{OT} \\ \alpha_{MR} \\ \alpha_{OR} \\ \alpha_{LF} \\ \alpha_{UN} \end{bmatrix}_t = x_t + \begin{bmatrix} 0 \\ c_{MR} \\ c_{OR} \\ c_{LF} \\ c_{UN} \end{bmatrix} + \begin{bmatrix} v_{OT} \\ v_{MR} \\ v_{OR} \\ v_{LF} \\ v_{UN} \end{bmatrix}_t \tag{21}$$

For each aging method $i$, we specified the following weakly informative priors

$$v_{i,t} \sim \text{Normal}(0, \sigma_{r,i}) \tag{22}$$

$$\sigma_{r,i} \sim \text{Half} - \text{normal}(0, 1) \tag{23}$$

Each model was run with four Markov Chain Monte-Carlo chains for 6000 iterations with an initial warmup phase of 3000 iterations. We visually inspected traceplots and ensured the chains were consistent with convergence for all estimated coefficients by checking that the potential scale reduction factor Rhat <1.01 and the effective sample size ESS > 1500 (Supplementary Data 1). Models were run in Stan[63] using the rstan v 2.21.8 package[64] in R v 4.2.0[65].

To assess whether changes in growth performance through time were due to species compositional changes, we used a Principal Coordinate Analysis (PCoA) and separated our observations into five distinct time slices: before 1940 (inclusive), 1941–1960, 1961–1980, 1981–2000, and after 2000. We used a Jaccard index based on presence/absence and visualised the results using a bivariate ordination plot. Although there were significant differences between the centroids of different time slices based on a PERMANOVA using 999 permutations ($p < 0.05$), this was driven by heterogeneity in group dispersion (i.e. time slices significantly differed in variance; $p < 0.05$). We ran the PCoA analysis using the vegan v 2.6–10 package[66].

## Phylogenetic analyses

To build a fish phylogeny, we followed the methods set forth by ref. 67. In short, we used the chronogram constructed by Rabosky et al. [68]. (www.fishtreeoflife.org), which has been time calibrated with the fossil record, as the backbone for the imputation of missing species based on taxonomy. We assigned taxonomic ranks for all species for which we have empirically measured growth performance values and applied the TACT stochastic polytomy resolution algorithm[69] to impute missing species onto the backbone tree. The TACT algorithm[69] uses models incorporating birth and death rates to calculate the diversification for taxonomic ranks. Species are then imputed within the most restrictive ranks according to their respective calculated diversification rates. We implemented the TACT algorithm using Python 3.12.7[70].

We used a Bayesian generalised mixed-effects model with a phylogenetic random effect to assess the interacting effects of fisheries management (i.e. managed, unmanaged, unfished) and sea surface temperature on the growth performance of marine fishes. The growth performance ($y_i$; where $i$ corresponds to a single population/stock) was modelled according to a gamma error distribution by specifying the gamma shape ($\tau$) and rate parameters ($\lambda_i$).

$$\text{Gamma}(y_i | \tau, \lambda_i) \tag{24}$$

$$\lambda_i = \frac{\tau}{\mu_i} \tag{25}$$

$$\begin{aligned} \log(\mu_i) = {} & \beta_0 + \beta_{fish}x_{fish} + \beta_{sst}x_{sst} + \beta_{int}x_{fish}x_{sst} \\ & + \alpha_{species} + \alpha_{method} + \alpha_{year} + b_{phylo} \end{aligned} \tag{26}$$

Here, $\beta_O$ is the overall intercept, $\beta_{fish}$ and $\beta_{sst}$ are the estimated effects of fisheries management and sea surface temperature, respectively; $\beta_{int}$ is the interaction term between fisheries management and sea surface temperature. We specified random intercepts for repeated species-level observations ($\alpha_{species}$), different aging methods ($\alpha_{method}$), and the sampling year ($\alpha_{year}$). The phylogenetic random effect is denoted with $b_{phylo}$. Because observations are taken at the species/stock level, the phylogenetic random effect $b_{phylo}$ accounts for the evolutionary non-independence among the residuals for interspecific contrasts (i.e. it allows for the fact that closely related species cannot be treated as independent samples; their evolutionary association makes them more likely to be similar). We specified the following weakly informative priors:

$$\lambda \sim \text{Gamma}(0.01, 0.01) \tag{27}$$

$$\beta_0 \sim \text{Normal}(0, 5) \tag{28}$$

$$\beta_{fish, sst, \text{int}} \sim \text{Normal}(0, 2) \tag{29}$$

We visually inspected traceplots and simulated residuals from the DHARMa v 0.4.6 package[71]. Models were run for 4000 iterations with an initial warmup phase of 2000 iterations using four Markov Chain Monte-Carlo chains. Phylogenetic models were run in R v 4.4.0[65] using the ape v 5.8[72] and brms v 2.21.0 packages[73].

### Reporting summary

Further information on research design is available in the Nature Portfolio Reporting Summary linked to this article.

## Data availability

The data analysed and produced in this study are available at Figshare (https://doi.org/10.6084/m9.figshare.28216268). There are no restrictions on data availability. Source data are provided with this paper.

## Code availability

The R code used to run analyses and generate figures is available at Figshare (https://doi.org/10.6084/m9.figshare.28216268).

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

## Acknowledgements

We are incredibly grateful for the statistical advice and support from Sean C. Anderson (and comments on earlier drafts) and Dan A. Greenberg; and James P.W. Robinson for valuable comments on previous drafts of this manuscript. We also thank the Australian Research Council (ARC: FL190100062, D.R.B.; DE250101047, A.C.S.) and an ARC Laureate PhD scholarship and a Natural Sciences and Engineering Research Council of Canada Postgraduate Doctoral Scholarship (HFY) for financial support.

## Author contributions

Helen F. Yan: Conceptualisation, formal analysis, methodology, data curation, investigation, visualisation, writing—original draft. Hannah V. Watkins: Formal analysis, methodology, writing—review and editing. Alexandre C. Siqueira: Methodology, data curation, writing—review and editing. David R. Bellwood: Writing—review and editing, supervision, funding acquisition.

## Competing interests

The authors declare no competing interests.
