## [Peer Review file · Nature Communications]

Over a century of global decline in the growth performance of marine fishes

Corresponding Author: Dr Helen Yan

Version 0:

Reviewer comments:

Reviewer #1

(Remarks to the Author)

This paper uses a very large database for its inference that the growth of exploited marine fishes has declined in the last decades and there is no arguing with this.

However, there are at least three ways that fish can and have become smaller, which are not really distinguished in this otherwise OK paper:

A) the first is that fishing tends to target the large individuals of a population. This leads to truncated size distributions and thus different growth parameters;

B) the second is that decade-long fishing as in A modifies the genetics of a population and induces a selection for small sizes. Fisheries scientists have real problems accepting this, but there is no reason to assume that evolutionary processes are different in fish than, e.g., in bighorn sheep, which hunters targeting the biggest-horned ones have turned into 'smallhorn sheep.' Indeed, there is a large body of recent literature on this negative selection in fishes;

C) fishes are getting smaller at higher temperature, which is described phenomenologically by Arkinson's 'temperature-size rule' (TSR) and mechanistically by, e.g., Pauly's gill-oxygen limitation theory.

Without an extensive section dealing and quantifying the possible contributions of A, B and C, the data presented in the paper cannot be properly evaluated, especially as the differentiation made now, between 'managed' and 'unmanaged' stocks are probably not as important as the authors think.

Reviewer #2

(Remarks to the Author)

I have already reviewed this study for Nature Ecology & Evolution about a year ago. My review for NEE was generally positive; I considered the study valuable and important, both for the results it provided and for the revised collection of life-history data (but see my comment below requesting more details on this). I therefore recommended that the study be published but asked for some clarifications regarding the methodology and for the authors to ensure that the code used for analyses was publicly available. My current view remains unchanged—it is a valuable contribution, but the authors should provide further methodological clarifications.

The current manuscript appears very similar to what I reviewed previously, but the authors now provide the code and include a section on modelling performance in a phylogenetic relatedness context. It is unfortunate that they did not provide a previous set of reviews and responses, as this would have made it easier to assess what has been changed and how previous comments were addressed. In general, such repeated submissions waste reviewers' and editors' time and should not be encouraged.

It is good to see that the authors are now providing access to the code through Figshare, but it does not appear to be complete, as it remains unclear how growth performance was calculated—something I specifically asked about in my previous review. Several other comments and questions from my earlier review were also not addressed and remain relevant (see below). Additionally, another reviewer from the NEE submission raised important concerns, which should be addressed or at least acknowledged in this manuscript. The main issue raised was confusion between intra- and interspecific processes: it is unclear whether growth performance declined within species or whether species with higher growth performance were replaced by sampling those with lower growth performance. I personally do not think that such systematic bias in sampling is likely. And if higher growth performance species were replaced with lower performance ones the final effect is the same - growth performance appears to be declining in the ocean. However, the authors must clarify this point explicitly in the discussion. They should also explain precisely what the phylogenetic models account for, as they do

not capture intra- versus interspecific contrasts. Finally, a previous reviewer could not understand why ageing method was included in the models. I believe it is essential to include this variable – the authors did the right thing but please add a couple of sentences clearly explaining this. Different ageing methods produce different L_{∞} and K values.

Relevant comments from my earlier review:

- The authors state that they individually assessed each reference of growth curves to check for errors. This would require reviewing thousands of publications (approximately 8,000 published curves in the final analysis) manually, which seems staggering. How was this accomplished? More importantly, what criteria were used to identify errors? Was the approach similar to that of Prince et al. (2023, ICES Journal of Marine Science), who proposed a specific methodology to assess the quality of published VB growth values available through FishBase? Which publications were identified as containing errors? This information alone would be an invaluable resource, as many researchers struggle with the low quality of published VB parameters on FishBase.

- Line 261: The conversion from fork/standard length to total length L_{∞} estimates is now clearer, but please provide the equations used for this transformation.

- Line 265: It is unclear how growth efficiency was estimated. A more detailed explanation of growth performance should be included in the main text. Please correct me if I am mistaken, but based on my understanding, the equation on line 272 would yield the following:

$$\phi = \log_{10}K + SL \times \log_{10}L_{\infty}$$

where $SL = -m \times b$, with $m = 0.76$ (line 253) and $b \approx 3$ (length-weight regression exponent).

If I use $K = 0.1$, $L_{\infty} = 100$, and $SL = -2.28$ (-0.76×3), I obtain $\phi = -5.58$.

If $K = 0.5$ and $L_{\infty} = 40$, then $\phi = -3.9$, and if K decreases to 0.4 (instead of 0.5), then $\phi = -4.05$.

However, the ϕ values reported in Fig. 3 range from 2.6 to 3.4, which I do not understand based on the provided methodology. The methods section does not allow these values to be reproduced.

Another point that was raised in the earlier review and which does not seem to be addressed is the fact that it is not clear whether the observed declines in growth performance over time are due to changes in performance within species or because species with lower growth performances were systematically sampled closer to present day (and higher growth performance species were sampled in the early 1900s. Some species were sampled multiple times, but the species composition through time does seem to differ. Without the information on when fish populations were sampled, it is impossible to disentangle whether growth performance has indeed declined with time due to fishing (as stated by the authors), or whether declines in growth performance are due to sampling different species at different time periods. In my view both aspects are interesting, but they should be clearly addressed.

Further, one reviewer in the previous review made a comment that this study is drawing intraspecific inferences from what are largely interspecific comparisons. The dataset combines measures from across many different species to draw its conclusions, i.e. the study is still drawing conclusions about changes within populations (due to evolution or plasticity) without actually examining changes in Φ through time within populations. The study also uses the interspecific scaling coefficient of 0.761 (Savage et al. 2004) to calculate the growth coefficient (Φ), which is a composite life history trait based on intraspecific measures of asymptotic size (L_{∞}) and the instantaneous rate at which an individual approaches its asymptotic size (K) – both parameters are intra-specific properties. Intraspecific metabolic scaling ranges from 0.40 to 1.29, and this aspect should be addressed and explored. Moreover, the study assumes a constant metabolic scaling exponent across the entire study period, which is not necessarily the case with fishing and warming. It is possible that L_{∞} , K , and metabolic scaling could all change over time so that Φ actually remains constant through time. Exploring how different values of L_{∞} , K , and metabolic scaling interact to produce Φ could be a useful exercise for exploring how robust Φ is to changes in L_{∞} , K , or metabolic scaling, or all three

Reviewer #3

(Remarks to the Author)

Dear authors and Editor,

Thank you very much for the opportunity to review “Over a century of global decline in growth performance of commercially valuable fishes”. I think it’s a fantastic manuscript that presents a much needed global-scale empirical assessment of trends in growth/size (there are a few model-based predictions). The data set collated is impressive and the modelling approach makes that justice. To understand how robust these conclusions are, I would like to see some additional explanations of the growth performance index, how composition of species/life histories has changed over time, and the assumption that managed species have had a higher exploitation history.

Here are my comments in order of appearance:

Title: Perhaps its somewhat of a missed opportunity to limit the title to “commercially valuable fishes”. I understand it’s because that’s where you saw the decline, however you’ve also looked a few non-commercial species, and I think that’s an important feature since most fishery data are biased towards commercial species. I don’t have any good suggestions though, but it there is one that keeps the title short and to the point it could be considered.

Line 24: Perhaps a detail, but are you looking at global impacts, or impacts globally? Since you extracted relatively “local” temperatures I’m thinking global is the scale of the analysis, not the impacts.

Line 28: You have a way of converting growth performance to size that I don’t really follow. I’ll get back to that. In this

summary section, I wonder though if you could specify which sizes we are talking about here that decline with 28%. Juvenile, maturation size, typical adult size, asymptotic size etc could all have different rates of declines for a given growth trajectory. Just thinking it's good to be specific! You cite Cheung et al 2013 for example, where the size decline is often misinterpreted.

Line 46: I think you can remove the "(but see13)" here, because you start the sentence by saying management can improve the status of fished stocks, then say however many are still overfished (and then again point to the paper showing managed stocks are in better shape).

Line 89: Can you elaborate on how the growth performance allows you to calculate a size and a growth rate? I don't really follow that step from the growth performance equation. At which ages are you looking? Related: on line 92 you indicate that the decline is likely because fish grow to smaller asymptotic sizes. Depending on how this translation works, could there be a situation where say young fish actually become bigger for a given age?

Line 72: I'm not very familiar with this growth index, but is this independent of life history strategy? You mention metabolic rate, for instance. I understand it corrects for mass, but are there still life history relationships in it? Are there systematic differences between say pelagic/demersal, mobile/passive species in this index? I'm curious because if this relationship exists it would be important to ensure that there is a consistency in which lifestyles / life history strategies are used over time.

Line 119: A key explanation of the results in this manuscript is that managed fisheries have a legacy of intense fishing. An argument is that species likely are managed because they have been exploited. However, 1) there are other factors influencing whether a species is managed (read: assessed), not least a regional pattern linked to how fisheries operate, how species-rich the fish community is, possibility to age easily with otoliths etc (Hilborn et al 2020). 2) effective management can improve stock status, and managed stocks have better status today, because they have appropriate fishing exploitation rates. Many un-managed stocks are in worse shapes because they experience too high fishing pressure. This also makes the heading questionable; is it really overfishing that drives these results, if managed stocks may be less overfished than fish-but-unmanaged stocks? I understand you argue there may be a delay here, but can you in some way show that fish in the category "managed fish" have had a higher cumulative fishing impact than fished-but-un-assessed fish? I think this is a critical assumption to support.

Line 127: Isn't approximately half of the catches from managed fisheries? Or am I misunderstanding the graph or the "well below those" part?

Line 140: Expresses => express?

Figure 4: consider some transformation of the x-axis. Perhaps a square-root transformation would be gentle enough to not distort the relationship but still allow to see how many observations you have for the unfished category.

Figure 4: Related to my previous comment; could you make a similar plot over time as well? How does the proportion temperate, subtropical and tropics change over time

Line 196: Can you put these estimates in a size context (which you so nicely did for the global trend). I get a feeling from looking at the predictions that these are very small! Given a change in the growth index of 0.025 when the temperature increases one degree (I hope that's a correct interpretation), how big effect does that have on the size?

Line 206: In this context, where you look at growth potential and not community reorganisation, I think it's important to state that approximately half of this 14-24 % decline in asymptotic size is due to physiology and the other half due to changes in species composition. In that sense it also better aligns with your findings.

Line 276: I'm a bit puzzled here. Do you use S_L as the slope (Fig. S1) or the anabolic term? Seems you used the average 0.761 value based on metabolic arguments, but why not the slope if you already have that available?

Line 322: I think here and possibly elsewhere it's important to be more explicit that you are looking at the effect of temperature over space, since each growth performance metric is paired with a semi-local temperature. I.e., as far as I understand, you are not looking at a given population with multiple observations through time, how warming has affected its growth potential. You mention "climate change" on line 25 for example, but this sort of analysis is if I'm correct not necessarily looking at climate warming, only under the assumption that spatial and temporal effects are similar.

Reviewer #4

(Remarks to the Author)

In the Anthropocene, the two greatest threats to ocean biodiversity are overexploitation and climate change, and these two factors are interacting. The impacts of overfishing are not clear, because impacts of climate change on the growth performance of fishes act as confounding factors. The authors directly assessed the combined impacts of fishing and temperature on growth performance, accounting for phylogenetic relatedness. They used a Bayesian generalised mixed-effects model with a phylogenetic random effect to assess these interacting factors.

The methods they used are sound, and the results are highly interesting and important. Therefore, I would like to recommend publication of paper.

It might be difficult for general readers to understand why phylogeny must be taken into account, and I suggest to add a short explanation for this point.

Journal name is lacking for Ref. 48 (White et al. 2022)
Science 2022 Aug 19;377(6608):834-839.

Version 1:

Reviewer comments:

Reviewer #2

(Remarks to the Author)

I have reviewed this manuscript twice already and I think it has improved substantially. I do not have any further comments, except for requesting the authors and editors to ensure that the compiled data and code are publicly available in a permanent repository. While the paper itself is a valuable and interesting contribution, the dataset of growth records will be used in many other future studies if it is made available.

All the best and congrats on a great study.

Asta Audzijonyte

Reviewer #3

(Remarks to the Author)

Please see attachement.

Reviewer #4

(Remarks to the Author)

I found, in the revised MS, my comments have been fully taken into account. Therefore, I recommend publication of this paper.

Reviewer #1 (Remarks to the Author):

This paper uses a very large database for its inference that the growth of exploited marine fishes has declined in the last decades and there is no arguing with this.

However, there are at least three ways that fish can and have become smaller, which are not really distinguished in this otherwise OK paper:

A) the first is that fishing tends to target the large individuals of a population. This leads to truncated size distributions and thus different growth parameters;

B) the second is that decade-long fishing as in A modifies the genetics of a population and induces a selection for small sizes. Fisheries scientists have real problems accepting this, but there is no reason to assume that evolutionary processes are different in fish than, e.g., in bighorn sheep, which hunters targeting the biggest-horned ones have turned into 'smallhorn sheep.' Indeed, there is a large body of recent literature on this negative selection in fishes;

C) fishes are getting smaller at higher temperature, which is described phenomenologically by Arkinson's 'temperature-size rule' (TSR) and mechanistically by, e.g., Pauly's gill-oxygen limitation theory.

Without an extensive section dealing and quantifying the possible contributions of A, B and C, the data presented in the paper cannot be properly evaluated, especially as the differentiation made now, between 'managed' and 'unmanaged' stocks are probably not as important as the authors think.

RESPONSE: We thank the reviewer for their comments and agree that these three mechanisms can simultaneously impact the growth performance of marine fishes. Unfortunately, we are unaware of any data that is currently available for which these mechanisms can be scrutinised and evaluated at global scales. To reconcile this, we have included additional text throughout the manuscript detailing each of the points. Responses to specific points are as follows:

A) Truncation of size distributions

We conducted additional multivariate analyses to show that the species composition through time have remained relative homogenous, highlighting that declines in growth performance are likely mainly driven by the size-selective nature of fisheries. This section reads as:

Declines in global growth performance can be indicative of both intraspecific changes and interspecific compositional changes. Despite considerable overlap in the families and species that were sampled through time, which suggests relative homogeneity in the annual species composition, there were also relatively distinct compositional shifts between specific time periods (Fig. S2). While it is plausible that declines in growth performance could be due to potential sampling biases (i.e. shifting from species with higher growth performance values towards those with lower growth performance through time), it is unlikely that this is the primary mechanism of the observed declines. Indeed, there is considerable overlap between the early (pre-1960) and late (post 2000) compositional assemblages, which have the most leverage on temporal trend estimates (Fig. S2). Additionally, species

compositions explained very little variation across growth performance studies (Fig. S2). It is therefore unlikely that the observed declines in growth performance were solely due to changes in the species sampled through time.

B) Fishing-induced evolutionary processes

We agree and have discussed how the truncation of populations can lead to the selection against larger body sizes. This reads as:

Therefore, the declines in growth performance of commercially valuable fishes could represent the interplay between two synergistic processes. First, an unfished, residual stock is remaining with smaller, younger individuals from size-selective fishing. Second, these demographic changes are then further reinforced and amplified by fisheries-induced selection against larger body sizes, which would manifest as lower growth performance values in subsequent generations⁴⁵⁻⁴⁸.

In the same aforementioned paragraph, we have now added additional text introducing the Rosa Lee phenomenon as a possible explanation for our results, which describes how the size-selective nature of fisheries can lead to differences in the expression of life-history traits in fishes. This section now reads as:

An alternative but not mutually exclusive mechanism for the observed declines in growth performance could be due to the “Rosa Lee” phenomenon⁴³. In short, larger, faster-growing individuals are targeted first by size-selective fisheries, allowing the smaller, slower-growing individuals of a population to survive⁴⁴. Consequently, for a given age, the remaining individuals are smaller and exhibit slower growth rates than would be expected in the absence of size-selective fishing⁴⁴, potentially resulting in declines in growth performance.

C) Temperature-size rule and GOLT

This is an interesting comment that points to a multifaceted problem. The relationship between temperature and growth is complex as it encompasses both physiological and ecological limitations. Although Pauly initially suggested that gills present a physiological limitation to growth, work by Bigman et al. (2023ab) has shown that gill surface area, an improved unbiased predictor compared to Pauly’s original gill area index, was only weakly related to growth and body size in fishes. The underlying mechanisms driving the temperature-size rule has been heavily debated, with multiple hypotheses (including Pauly’s GOLT and the metabolic theory of ecology) presented as co-occurring mechanisms that are not mutually exclusive (reviewed by Audzijonyte et al. 2019). This is a complex debate and outside the scope of our current study, but presents a very fruitful research avenue for the future.

References

Audzijonyte, A., Barneche, D. R., Baudron, A. R., Belmaker, J., Clark, T. D., Marshall, C. T., ... & van Rijn, I. (2019). Is oxygen limitation in warming waters a valid mechanism to explain decreased body sizes in aquatic ectotherms? *Global Ecology and Biogeography*, 28(2), 64-77.

Bigman, J. S., Wegner, N. C., & Dulvy, N. K. (2023a). Gills, growth and activity across fishes. *Fish and Fisheries*, 24(5), 730-743.

Bigman, J. S., Wegner, N. C., & Dulvy, N. K. (2023b). Revisiting a central prediction of the Gill Oxygen Limitation Theory: Gill area index and growth performance. *Fish and Fisheries*, 24(3), 354-366.

Reviewer #2 (Remarks to the Author):

I have already reviewed this study for *Nature Ecology & Evolution* about a year ago. My review for NEE was generally positive; I considered the study valuable and important, both for the results it provided and for the revised collection of life-history data (but see my comment below requesting more details on this). I therefore recommended that the study be published but asked for some clarifications regarding the methodology and for the authors to ensure that the code used for analyses was publicly available. My current view remains unchanged—it is a valuable contribution, but the authors should provide further methodological clarifications.

RESPONSE: Thank you for your attention with our manuscript, we greatly appreciate your comments and repeated assessment of our work.

The current manuscript appears very similar to what I reviewed previously, but the authors now provide the code and include a section on modelling performance in a phylogenetic relatedness context. It is unfortunate that they did not provide a previous set of reviews and responses, as this would have made it easier to assess what has been changed and how previous comments were addressed. In general, such repeated submissions waste reviewers' and editors' time and should not be encouraged.

RESPONSE: We agree and apologise. We have now incorporated all of the reviewer's comments and believe that they have greatly improved our manuscript.

It is good to see that the authors are now providing access to the code through Figshare, but it does not appear to be complete, as it remains unclear how growth performance was calculated—something I specifically asked about in my previous review.

RESPONSE: We apologise for the oversight and have updated the scripts available to include the lines of code showing how growth performance was calculated.

Several other comments and questions from my earlier review were also not addressed and remain relevant (see below). Additionally, another reviewer from the NEE submission raised important concerns, which should be addressed or at least acknowledged in this manuscript. The main issue raised was confusion between intra- and interspecific processes: it is unclear whether growth performance declined within species or whether species with higher growth performance were replaced by sampling those with lower growth performance. I personally do not think that such systematic bias in sampling is likely. And if higher growth performance species were replaced with lower performance ones the final effect is the same - growth performance appears to be declining in the ocean. However, the authors must clarify this point explicitly in the discussion.

RESPONSE: This is a good point, thank you for bringing it to our attention again. While both mechanisms are plausible (i.e. demographic changes in the size/age distributions of fishes versus changes in the species that are sampled through time), we initially controlled for the latter by including a family-level grouping factor in our models, which accounts for repeated sampling within families. This random effect term means that the global mean for each year can be interpreted as the expected mean growth performance if all families were sampled in that year. Provided there is at least some continuity in the families observed from year-to-year (i.e. there is not a completely new set of families included in later years than in earlier years), this modelling approach can resolve issues related to biased sampling over time (e.g. if families with lower growth performances were oversampled in later years). Importantly, there are many families that are present throughout the entirety of the time series (e.g. Clupeidae, Pleuronectidae, Labridae). Additionally, each of these families comprise species that are also present throughout the entire time series (e.g. *Sprattus sprattus*, *Pleuronectes platessa*, *Ctenolabrus rupestris*), indicating that there is also some compositional homogeneity within families through time and further reinforcing our confidence that our results are not being unduly influenced by compositional changes.

However, to more directly explore this issue, we now further explore changes in species sampled through time using Jaccard distances of the species per year to show how years clustered in multivariate space using a Principal Coordinate Analysis. In short, while there are some time periods that are relatively distinct from others, there was considerable overlap as well (as denoted by their respective 90% ellipses; Fig. S2), which suggests that species compositions have not markedly or consistently shifted through time. Notably, the median annual growth performance did not display any obvious clustering in multivariate space, with some of the lowest recorded values found in earlier time slices. If declines in growth performance were due to changes in species composition alone, we would expect to find distinct shifts in median annual growth performance values to match the clusters of each time slice. Therefore, the observed declines in growth performance are unlikely to be due primarily to changes in species composition through time. To address this, we have added a paragraph to the discussion, additional text to the methods, and a new supplemental figure (Fig. S2). The new discussion paragraph reads as:

Declines in global growth performance can be indicative of both intraspecific changes and interspecific compositional changes. Despite considerable overlap in the families and species that were sampled through time, which suggests relative homogeneity in the annual species composition, there were also relatively distinct compositional shifts between specific time periods (Fig. S2). While it is plausible that declines in growth performance could be due to potential sampling biases (i.e. shifting from species with higher growth performance values towards those with lower growth performance through time), it is unlikely that this is the primary mechanism of the observed declines. Indeed, there is considerable overlap between the early (pre-1960) and late (post 2000) compositional assemblages, which have the most leverage on temporal trend estimates (Fig. S2). Additionally, species compositions explained very little variation across growth performance studies (Fig. S2). It is therefore unlikely that the observed declines in growth performance were solely due to changes in the species sampled through time. Irrespective of the underlying mechanism, given the inherent temporal lag between management implementation and stock recovery¹², the legacy of size-based fishing could manifest throughout populations as declines in growth performance, even when abundances are stable or increasing. Regardless of whether overfishing is the proximate or distal cause for the decline in fish growth performance, changes in the life histories of fishes

are likely to be slow to reverse (or potentially irreversible) and can have repercussions on the biomass and productivity of capture fisheries^{45,47}.

They should also explain precisely what the phylogenetic models account for, as they do not capture intra- versus interspecific contrasts.

RESPONSE: The phylogeny accounts for the evolutionary non-independence among the residuals of the generalised linear mixed-effects model at the species level and therefore captures interspecific contrasts. We have now added text to the Methods explicitly stating this, which now reads as:

Because observations are taken at the species/stock level, the phylogenetic random effect b_{phylo} accounts for the evolutionary non-independence among the residuals for interspecific contrasts (i.e. it allows for the fact that closely related species cannot be treated as independent samples; their evolutionary association makes them more likely to be similar).

Finally, a previous reviewer could not understand why ageing method was included in the models. I believe it is essential to include this variable – the authors did the right thing but please add a couple of sentences clearly explaining this. Different ageing methods produce different L_{∞} and K values.

RESPONSE: We have now added more explicit text to the methods, which reads as:

Individual aging methods (i.e. mark recapture, otolith rings, other rings (e.g. scales, vertebrae), length frequency analyses, and unknown) each contain their own mechanistic biases, which can lead to different values of L_{∞} and K (and thus, different growth performance values ϕ)^{58,62}. Indeed, aging method accounts for roughly 5% of the variation in growth rates across fishes²⁸. To account for these inherent methodological biases quantitatively, each method was modelled with its own respective observation deviance.

Relevant comments from my earlier review:

- The authors state that they individually assessed each reference of growth curves to check for errors. This would require reviewing thousands of publications (approximately 8,000 published curves in the final analysis) manually, which seems staggering. How was this accomplished? More importantly, what criteria were used to identify errors? Was the approach similar to that of Prince et al. (2023, ICES Journal of Marine Science), who proposed a specific methodology to assess the quality of published VB growth values available through FishBase? Which publications were identified as containing errors? This information alone would be an invaluable resource, as many researchers struggle with the low quality of published VB parameters on FishBase.

RESPONSE: Collating this dataset was a dominant component of the lead author's doctoral thesis work. Indeed, she spent almost a year going through the publications. This work has been used in a recently published article (ref. 28 in the main text). The original search yielded 141,649 articles, which were manually evaluated following the steps outlined in the first paragraph of the Methods. Regrettably, we did not record which growth curves from FishBase contained errors, but they were mainly typological or rounding errors.

- Line 261: The conversion from fork/standard length to total length L_{∞} estimates is now clearer, but please provide the equations used for this transformation.

RESPONSE: We have now included the equation in the methods. This section reads as:

To make studies comparable, we used published length-length conversion coefficients from FishBase⁵⁵ to convert estimates of L_∞ recorded as standard or fork length to total length in cm using the following equation:

$$\text{Total length} = a + b \times \text{standard or fork length}$$

Here, a and b are the length-length conversion factors from FishBase⁵⁵. When species-level coefficients were not available, we used genus or family means.

- Line 265: It is unclear how growth efficiency was estimated. A more detailed explanation of growth performance should be included in the main text. Please correct me if I am mistaken, but based on my understanding, the equation on line 272 would yield the following:

$$\phi = \log_{10}K + SL \times \log_{10}L_\infty$$

where $SL = -m \times b$, with $m = 0.76$ (line 253) and $b \approx 3$ (length-weight regression exponent). If I use $K = 0.1$, $L_\infty = 100$, and $SL = -2.28$ (-0.76×3), I obtain $\phi = -5.58$. If $K = 0.5$ and $L_\infty = 40$, then $\phi = -3.9$, and if K decreases to 0.4 (instead of 0.5), then $\phi = -4.05$. However, the ϕ values reported in Fig. 3 range from 2.6 to 3.4, which I do not understand based on the provided methodology. The methods section does not allow these values to be reproduced.

RESPONSE: Thank you for bringing this technical misspecification to our attention. You are correct that S_L is generally negative because it describes the slope between $\log_{10}K$ and $\log_{10}L_\infty$. The double-log₁₀ relationship can be represented by the following equation:

$$\log_{10}K = S_L \times \log_{10}L_\infty + \beta \quad (\text{R1})$$

Here, β is the general intercept. Growth performance (ϕ) is the specific β calculated where $\log_{10}L_\infty = 0$ (i.e. $L_\infty = 1$ cm). Rearranging the formula to solve for the intercept ϕ produces:

$$\phi = \log_{10}K - S_L \times \log_{10}L_\infty \quad (\text{R2})$$

We attempted to simplify the equation due to the double negative produced by the subtraction and the negative S_L slope. However, you are correct that this is inaccurate because S_L is generally negative, so we have rewritten the formula in the main text to reflect equation (R2), which matches the code that is now provided. With these corrections, your theoretical examples would produce $\phi = 3.56$ and 3.35, respectively, which are within the range of our observed ϕ values.

Another point that was raised in the earlier review and which does not seem to be addressed is the fact that it is not clear whether the observed declines in growth performance over time are due to changes in performance within species or because species with lower growth performances were systematically sampled closer to present day (and higher growth performance species were sampled in the early 1900s. Some species were sampled multiple times, but the species composition through time does seem to differ. Without the information on when fish populations were sampled, it is impossible to disentangle whether growth performance has indeed declined with time due to fishing (as stated by the authors), or whether declines in growth performance are due to sampling different species at different time periods. In my view both aspects are interesting, but they should be clearly addressed.

RESPONSE: This has now been addressed. Please see our specific responses above.

Further, one reviewer in the previous review made a comment that this study is drawing intraspecific inferences from what are largely interspecific comparisons. The dataset combines measures from across many different species to draw its conclusions, i.e. the study is still drawing conclusions about changes within populations (due to evolution or plasticity) without actually examining changes in Φ through time within populations. The study also uses the interspecific scaling coefficient of 0.761 (Savage et al. 2004) to calculate the growth coefficient (Φ), which is a composite life history trait based on intraspecific measures of asymptotic size (L_∞) and the instantaneous rate at which an individual approaches its asymptotic size (K) – both parameters are intra-specific properties. Intraspecific metabolic scaling ranges from 0.40 to 1.29, and this aspect should be addressed and explored. Moreover, the study assumes a constant metabolic scaling exponent across the entire study period, which is not necessarily the case with fishing and warming. It is possible that L_∞ , K , and metabolic scaling could all change over time so that Φ actually remains constant through time. Exploring how different values of L_∞ , K , and metabolic scaling interact to produce Φ could be a useful exercise for exploring how robust Φ is to changes in L_∞ , K , or metabolic scaling, or all three.

RESPONSE: This is a good point, thank you for carrying it over from our previous submission. We have now re-run the global time series model but with varying S_L values. In short, we calculated S_L for all species that had at least three growth curves – we could not calculate S_L per species within each year because there were not enough repeated observations per method. Mainly, we still find evidence of a decline in growth performance throughout the time series (now shown in Figure S8). We note, however, that varying S_L at the species level produced more extreme values of growth performance than using the constant scaling exponent from Savage et al. (2004), which is now shown in Figure S7. Indeed, the values of growth performance produced exceeded both the upper and lower bounds of both Pauly (1979) and Morais and Bellwood (2018), who also calculated interspecific growth performance values (note that growth performance from Pauly (1979) is based on W_∞ instead of L_∞).

Source	Lower limit ϕ	Upper limit ϕ
Pauly (1979)	-0.7	6.20
Morais and Bellwood (2018)	1.5	4.85
This paper – constant S_L	0.087	6.5
This paper – varying S_L	-15.43	41.07

These extreme growth performance values are likely produced by the high leverage of individual growth curves, which can be affected by methodological biases (e.g. underrepresentation of body size ranges) or environmental factors. These factors that are methodologically difficult to account for likely explain why previous studies have chosen to use a common S_L value to normalise extreme observations towards a unified auximetric relationship, grounded in metabolic theory (Morais and Bellwood 2018; Pauly 1979). Additionally, the time series using a variable S_L resulted in a larger magnitude of decline with greater uncertainty: 21% [90% credible interval: -39.3% to 1.53%] decline compared to 7.9% [4.4% to 11.4%] using a constant S_L . We have therefore retained the conservative analyses using a “constant” S_L in the main text (i.e. the product of the anabolic term and the length-weight regression exponent) but present the global time series using variable S_L values in the

supplementary materials (Fig. S8). These changes have been detailed in the Methods, which reads as:

To test the sensitivity of ϕ to different S_L values, we re-ran the global time series model but calculated ϕ using empirically derived S_L values. We calculated S_L by regressing $\log_{10}(K)$ against $\log_{10}(L_\infty)$ for all species-level observations that had more than three observations per aging method (adapted from ref.^{32,58}). This resulted in 714 species with empirically measured S_L values. We used the “constant” S_L value for all species without empirically measured estimates by multiplying the mean anabolic term by the length-weight regression exponent (as denoted by equation 3). While we found a relatively positive association between both methods of calculating growth performance (i.e. constant vs. varying S_L values; Fig. S7), using varying S_L values produced much more extreme values of growth performance compared to using constant values (-15.43 to 41.07 vs. 0.087 to 6.5, respectively), which exceeded the estimates of previous interspecific growth studies^{58,60}. Despite these extreme values, we found a steeper, albeit more uncertain pattern of decline in the global time series (median decline [90% credible interval]: 21.0% [-39.3% to 1.53%]; Fig. S8). We therefore present the conservative estimates generated using the “constant” S_L values in the main text. See supplementary methods for model specifications.

References

- Morais, R. A., & Bellwood, D. R. (2018). Global drivers of reef fish growth. *Fish and Fisheries*, 19(5), 874-889.
- Pauly, D. (1979). Gill size and temperature as governing factors in fish growth: A generalization of von Bertalanffy's growth formula. *Berichte aus dem Institut für Meereskunde Kiel*, 63, 1-156.
- Savage, V. M., Gillooly, J. F., Brown, J. H., West, G. B., & Charnov, E. L. (2004). Effects of body size and temperature on population growth. *The American Naturalist*, 163(3), 429-441.

Reviewer #3 (Remarks to the Author):

Dear authors and Editor,

Thank you very much for the opportunity to review “Over a century of global decline in growth performance of commercially valuable fishes”. I think it’s a fantastic manuscript that presents a much needed global-scale empirical assessment of trends in growth/size (there are a few model-based predictions). The data set collated is impressive and the modelling approach makes that justice. To understand how robust these conclusions are, I would like to see some additional explanations of the growth performance index, how composition of species/life histories has changed over time, and the assumption that managed species have had a higher exploitation history.

RESPONSE: Thank you for your constructive and positive feedback on our manuscript. We greatly appreciate your suggestions and believe that they have strengthened our work.

Here are my comments in order of appearance:

Title: Perhaps its somewhat of a missed opportunity to limit the title to “commercially valuable fishes”. I understand it’s because that’s where you saw the decline, however you’ve also looked a few non-commercial species, and I think that’s an important feature since most fishery data are biased towards commercial species. I don’t have any good suggestions though, but it there is one that keeps the title short and to the point it could be considered.

RESPONSE: In light of the comments that you have provided below, we agree and have now changed the title to exclude commercially valuable fishes. It now reads: *Over a century of global decline in the growth performance of marine fishes.*

Line 24: Perhaps a detail, but are you looking at global impacts, or impacts globally? Since you extracted relatively “local” temperatures I’m thinking global is the scale of the analysis, not the impacts.

RESPONSE: We have changed the text accordingly. This sentence now reads as:

Here, we assessed the impacts of fishing and temperature on the growth performance of marine teleost fishes globally over the last century.

Line 28: You have a way of converting growth performance to size that I don’t really follow. I’ll get back to that. In this summary section, I wonder though if you could specify which sizes we are talking about here that decline with 28%. Juvenile, maturation size, typical adult size, asymptotic size etc could all have different rates of declines for a given growth trajectory. Just thinking it’s good to be specific! You cite Cheung et al 2013 for example, where the size decline is often mis-interpreted.

RESPONSE: Thank you for highlighting our need for clarity. We previously extrapolated body sizes and growth trajectories from the raw data based on the range of modelled growth performance values, which was likely to lead to confusion. We have now replaced these estimates with modelled values. Because growth performance is intrinsically capturing the trade-off between L_{∞} and K , we converted changes in growth performance into changes in *either* L_{∞} or K by holding the other parameter constant (i.e. holding K and L_{∞} constant, respectively). We have added extra text and the mathematical formulations of these calculations in the Methods, which reads as:

We used equation (2) to generate hypothetical scenarios depicting how changes in growth performance ($\Delta\phi$) across time steps (t) would result in changes in either L_{∞} and K (while holding K and L_{∞} constant, respectively). Here, we used the “constant” S_L value (see above) with the mean length-weight regression exponent across species ($b = 3.03$).

$$\Delta\phi = \phi_t - \phi_{t-1} = [\log_{10}K_t - S_L \times \log_{10}L_{\infty,t}] - [\log_{10}K_{t-1} - S_L \times \log_{10}L_{\infty,t-1}] \quad (4)$$

$$\Delta\phi = \log_{10}\left(\frac{K_t}{K_{t-1}}\right) - S_L \times \log_{10}\left(\frac{L_{\infty,t}}{L_{\infty,t-1}}\right) \quad (4.1)$$

Solving equation (4.1) to find changes in K while keeping L_{∞} constant removes the entire L_{∞} term by setting $S_L \times \log_{10}(1) = 0$.

$$\left(\frac{K_t}{K_{t-1}}\right) = 10^{\Delta\phi} \quad (4.2)$$

Conversely, solving equation (4.1) to find changes in L_∞ produces:

$$\left(\frac{L_{\infty,t}}{L_{\infty,t-1}} \right) = 10^{\frac{-\Delta\phi}{S_L}} \quad (4.3)$$

Because these are modelled estimates based on general patterns, we are unable to specify the ontogenetic stage for which these changes are occurring. Please note that we have removed the citations from the Abstract to match the journal style.

Line 46: I think you can remove the “(but see13)” here, because you start the sentence by saying management can improve the status of fished stocks, then say however many are still overfished (and then again point to the paper showing managed stocks are in better shape).

RESPONSE: Done.

Line 89: Can you elaborate on how the growth performance allows you to calculate a size and a growth rate? I don't really follow that step from the growth performance equation. At which ages are you looking? Related: on line 92 you indicate that the decline is likely because fish grow to smaller asymptotic sizes. Depending on how this translation works, could there be a situation where say young fish actually become bigger for a given age?

RESPONSE: This is now clarified. Please see specific response above.

Line 72: I'm not very familiar with this growth index, but is this independent of life history strategy? You mention metabolic rate, for instance. I understand it corrects for mass, but are there still life history relationships in it? Are there systematic differences between say pelagic/demersal, mobile/passive species in this index? I'm curious because if this relationship exists it would be important to ensure that there is a consistency in which lifestyles / life history strategies are used over time.

RESPONSE: We have corrected this sentence to indicate that growth performance standardises for body *size* not *mass*. This sentence now reads as:

Specifically, growth performance standardises estimates of instantaneous growth rates against a single unit of body size³³.

Exploring differences in growth performance across different ecological traits and/or lifestyles would be very interesting; unfortunately, very few interspecific studies have been conducted across marine fishes outside aquaculture or laboratory settings. For example, Morais and Bellwood (2018) showed that K_{max} , another standardisation of growth in marine fishes, was marginally affected by ecological traits, such as position in the water column. The uncorrelated nature of K_{max} and growth performance (see Fig. 4b from Morais and Bellwood 2018) however, prohibits the extrapolation of a priori expectations across metrics. The main studies that evaluate growth performance in interspecific contexts are relating growth to metabolic rate (e.g. Wong et al. 2021) or proxies (e.g. gill surface area; Bigman et al., 2023; Pauly 2021). Exploring systematic differences between ecological lifestyles is outside the scope of the work herein, but would be a very interesting avenue of research to explore.

References

Bigman, J. S., Wegner, N. C., & Dulvy, N. K. (2023). Revisiting a central prediction of the Gill Oxygen Limitation Theory: Gill area index and growth performance. *Fish and Fisheries*, 24(3), 354-366.

Pauly, D. (2021). The gill-oxygen limitation theory (GOLT) and its critics. *Science Advances*, 7(2), eabc6050.

Wong, S., Bigman, J. S., & Dulvy, N. K. (2021). The metabolic pace of life histories across fishes. *Proceedings of the Royal Society B*, 288(1953), 20210910.

Line 119: A key explanation of the results in this manuscript is that managed fisheries have a legacy of intense fishing. An argument is that species likely are managed because they have been exploited. However, 1) there are other factors influencing whether a species is managed (read: assessed), not least a regional pattern linked to how fisheries operate, how species-rich the fish community is, possibility to age easily with otoliths etc (Hilborn et al 2020). 2) effective management can improve stock status, and managed stocks have better status today, because they have appropriate fishing exploitation rates. Many un-managed stocks are in worse shapes because they experience too high fishing pressure. This also makes the heading questionable; is it really overfishing that drives these results, if managed stocks may be less overfished than fish-but-unmanaged stocks? I understand you argue there may be a delay here, but can you in some way show that fish in the category “managed fish” have had a higher cumulative fishing impact than fished-but-un-assessed fish? I think this is a critical assumption to support.

RESPONSE: This is a really good point, thank you for bringing it up. Given the lack of management support for unassessed fishes, we are unlikely able to empirically capture the full breadth of fisheries exploitation that this group of fishes has received relative to managed fishes. As illustrated by Fig. 4b, the majority of unmanaged fishes are concentrated in the warm-water, tropical regions. The tropics are typically dominated by small-scale, coral reef fisheries that are reliant on locally managed fisheries, which are often controlled by local communities rather than federal agencies. While local management practices can be incredibly effective (e.g. Cinner et al. 2019), they are also misrepresented in national census measures and typically lack the financial resources to conduct formal stock assessments (Pauly et al. 2013). Due to these inherent biases in global fisheries data, it is possible that the steady level of growth performance that we observed was driven by two likely co-occurring mechanisms: (1) growth studies on unmanaged species are underprioritised by scientists and are thus inadequately represented in the current growth literature and/or (2) many fisheries in these regions have been severely fished to the point that they are already operating on a shifted baseline of generally lower growth performance. Although we are unable to distil which mechanism is driving our results with the data at hand, we have included an additional discussion paragraph highlighting these biases, which reads as:

Unmanaged fisheries are typically exposed to unsustainable fishing pressures, therefore we might expect that growth performance would follow, if not surpass, the same magnitude of decline found in commercially valuable fishes. However, we found that the growth performance of unmanaged fishes was relatively stable throughout the time series. Although an underwhelming pattern of stagnation could be indicative of resilient demographic processes (i.e. resilience to fishing and temperature changes) or lagged responses, it is also possible that the focus on unmanaged fishes is disproportionately higher in tropical regions and may have biased the current literature on growth across marine fishes. The tropics are

dominated by small-scale coral reef fisheries, where management is typically regulated by local communities rather than national-level policies⁵³. Consequently, many coral reef nations lack the resources or funding to conduct stock assessments⁵⁴ and such unmanaged stocks are thus misrepresented in global fisheries census data¹⁰. Given that growth studies can be tightly linked to conventional fisheries management practices¹³, the lack of stock assessments in the tropics may have led to an under-representation of unmanaged fishes in the growth literature. Indeed, unmanaged fishes may already be operating on a shifted baseline, whereby stocks are severely depleted to the point of mainly comprising individuals with lower growth performance. Care is needed, therefore, as these systematic biases could distort or completely omit any realised changes in growth performance across unmanaged fishes.

References

Cinner, J. E., Lau, J. D., Bauman, A. G., Feary, D. A., Januchowski-Hartley, F. A., Rojas, C. A., ... & Graham, N. A. J. (2019). Sixteen years of social and ecological dynamics reveal challenges and opportunities for adaptive management in sustaining the commons. *Proceedings of the National Academy of Sciences*, 116(52), 26474-26483.

Pauly, D., Hilborn, R., & Branch, T. A. (2013). Fisheries: Does catch reflect abundance? *Nature*, 494(7437), 303-306.

Line 127: Isn't approximately half of the catches from managed fisheries? Or am I misunderstanding the graph or the "well below those" part?

RESPONSE: You are correct, thank you for pointing that out. We have modified the text to read:

Although the catches of unmanaged fishes followed a similar pattern to managed fishes, the catch of unmanaged fisheries is approximately half that of managed fisheries³⁵ (Fig. S3).

Line 140: Expresses => express?

RESPONSE: Changed accordingly.

Figure 4: consider some transformation of the x-axis. Perhaps a square-root transformation would be gentle enough to not distort the relationship but still allow to see how many observations you have for the unfished category.

RESPONSE: We have recreated the figure using the square root transformation as suggested. However, we found the distortion of the x axis reduced the interpretability of the values considerably, while using another transformation (i.e. log) distorted the relationship too drastically. We therefore chose to keep the original scaling in the main text but have provided a supplemental figure with a square root transformation in the supplementary materials (Fig. S4).

Figure 4: Related to my previous comment; could you make a similar plot over time as well? How does the proportion temperate, subtropical and tropics change over time

RESPONSE: We have now added a supplemental figure (Fig. S9) showing the changing proportions through time and added text to the Methods highlighting this new figure. The text reads as:

Across all three regions, the number of observations noticeably increased from the 1950s to the early 2000s, before steadily decreasing after 2010 (Fig. S9).

Line 196: Can you put these estimates in a size context (which you so nicely did for the global trend). I get a feeling from looking at the predictions that these are very small! Given a change in the growth index of 0.025 when the temperature increases one degree (I hope that's a correct interpretation), how big effect does that have on the size?

RESPONSE: Although we agree that visualising the size equivalence of these modelled results would be useful, the structure of this model means that the effect of a single degree is different depending on its value. The phylogenetic models use a gamma error distribution and the estimates provided are based on a log link, meaning that the effect of temperature on growth performance is exponential and a one-degree difference will have different effects along the range of temperatures. We therefore followed a similar formulation produced by equations 4-4.3 and calculated the proportional changes in L_{∞} and K with a 1°C change in the lower and upper ranges of our data. This section now reads as:

For unmanaged fishes, a 1°C increase in temperature translated to, at most, a 7-15% median increase in L_{∞} or an 18-37% increase in K across the majority of the range of observed temperatures (smaller estimates calculated from 0-1°C and larger estimates from 28-29°C). For unfished species, a similar increase in temperature resulted in a maximum increase of 14-48% in L_{∞} or a 34-147% increase in K .

Line 206: In this context, where you look at growth potential and not community reorganisation, I think it's important to state that approximately half of this 14-24 % decline in asymptotic size is due to physiology and the other half due to changes in species composition. In that sense it also better aligns with your findings.

RESPONSE: Done. This sentence now reads as:

Given the inherent correlation between metabolic rates and growth^{30,51}, models of energy assimilation predict that under a high-emissions climate scenario, the asymptotic body sizes of fishes may decrease by 14-24% globally in the next 30 years due to changes in both physiology and species assemblages²⁵.

Line 276: I'm a bit puzzled here. Do you use S_L as the slope (Fig. S1) or the anabolic term? Seems you used the average 0.761 value based on metabolic arguments, but why not the slope if you already have that available?

RESPONSE: We apologise for the confusion. Based on Pauly (1979) and Morais and Bellwood (2018), we used S_L as the slope, but it is calculated based on the anabolic term (0.761) from Savage et al. (2004) using the following equation: $S_L = -m * b$ where m is the anabolic term and b is the length-weight regression parameter (now Equation 3 in the main text).

We have re-run the global time series model but with varying S_L values by regressing $\log_{10}(K)$ against $\log_{10}(L_\infty)$. In short, we calculated S_L for all species that had at least three growth curves – we could not calculate S_L per species within each year because there were not enough repeated observations per method. Mainly, we still find evidence of a decline in growth performance throughout the time series (now shown in Figure S8). We note, however, that varying S_L at the species level produced more extreme values of growth performance than using the constant scaling exponent from Savage et al. (2004), which is now shown in Figure S7. Indeed, the values of growth performance produced exceeded both the upper and lower bounds of both Pauly (1979) and Morais and Bellwood (2018), who also calculated interspecific growth performance values (note that growth performance from Pauly (1979) is based on W_∞ instead of L_∞).

Source	Lower limit ϕ	Upper limit ϕ
Pauly (1979)	-0.7	6.20
Morais and Bellwood (2018)	1.5	4.85
This paper – constant S_L	0.087	6.5
This paper – varying S_L	-15.43	41.07

These extreme growth performance values are likely produced by the high leverage of individual growth curves, which can be affected by methodological biases (e.g. underrepresentation of body size ranges) or environmental factors. These factors that are methodologically difficult to account for likely explain why previous studies have chosen to use a common S_L value to normalise extreme observations towards a unified auximetric relationship, grounded in metabolic theory (Morais and Bellwood 2018; Pauly 1979). Additionally, the time series using a variable S_L resulted in a larger magnitude of decline with greater uncertainty: 21% [90% credible interval: -39.3% to 1.53%] decline compared to 7.9% [4.4% to 11.4%] using a constant S_L . We have therefore retained the conservative analyses using a common S_L in the main text but present the global time series using variable S_L values in the supplementary materials (Fig. S8). These changes have been detailed in the Methods, which reads as:

To test the sensitivity of ϕ to different S_L values, we re-ran the global time series model but calculated ϕ using empirically derived S_L values. We calculated S_L by regressing $\log_{10}(K)$ against $\log_{10}(L_\infty)$ for all species-level observations that had more than three observations per aging method (adapted from ref.^{32,58}). This resulted in 714 species with empirically measured S_L values. We used the “constant” S_L value for all species without empirically measured estimates by multiplying the mean anabolic term by the length-weight regression exponent (as denoted by equation 3). While we found a relatively positive association between both methods of calculating growth performance (i.e. constant vs. varying S_L values; Fig. S7), using varying S_L values produced much more extreme values of growth performance compared to using constant values (-15.43 to 41.07 vs. 0.087 to 6.5, respectively), which exceeded the estimates of previous interspecific growth studies^{58,60}. Despite these extreme values, we found a steeper, albeit more uncertain pattern of decline in the global time series (median decline [90% credible interval]: 21.0% [-39.3% to 1.53%]; Fig. S8). We therefore present the conservative estimates generated using the “constant” S_L values in the main text. See supplementary methods for model specifications.

References

Morais, R. A., & Bellwood, D. R. (2018). Global drivers of reef fish growth. *Fish and Fisheries*, 19(5), 874-889.

Pauly, D. (1979). Gill size and temperature as governing factors in fish growth: A generalization of von Bertalanffy's growth formula. *Berichte aus dem Institut für Meereskunde Kiel*, 63, 1-156.

Savage, V. M., Gillooly, J. F., Brown, J. H., West, G. B., & Charnov, E. L. (2004). Effects of body size and temperature on population growth. *The American Naturalist*, 163(3), 429-441.

Line 322: I think here and possibly elsewhere it's important to be more explicit that you are looking at the effect of temperature over space, since each growth performance metric is paired with a semi-local temperature. I.e., as far as I understand, you are not looking at a given population with multiple observations through time, how warming has affected its growth potential. You mention "climate change" on line 25 for example, but this sort of analysis is if I'm correct not necessarily looking at climate warming, only under the assumption that spatial and temporal effects are similar.

RESPONSE: This is a good point, thank you for bringing it up. We have specified throughout the text that we are evaluating the effect of temperature over space.

Reviewer #4 (Remarks to the Author):

In the Anthropocene, the two greatest threats to ocean biodiversity are overexploitation and climate change, and these two factors are interacting. The impacts of overfishing are not clear, because impacts of climate change on the growth performance of fishes act as confounding factors. The authors directly assessed the combined impacts of fishing and temperature on growth performance, accounting for phylogenetic relatedness. They used a Bayesian generalised mixed-effects model with a phylogenetic random effect to assess these interacting factors.

The methods they used are sound, and the results are highly interesting and important. Therefore, I would like to recommend publication of paper.

RESPONSE: Thank you for your positive evaluation of our work.

It might be difficult for general readers to understand why phylogeny must be taken into account, and I suggest to add a short explanation for this point.

RESPONSE: We have added additional text to the Methods describing why we included a phylogenetic random effect. This new sentence reads as:

Because observations are taken at the species/stock level, the phylogenetic random effect b_{phylo} accounts for the evolutionary non-independence among the residuals for interspecific contrasts (i.e. it allows for the fact that closely related species cannot be treated as independent samples; their evolutionary association makes them more likely to be similar).

Journal name is lacking for Ref. 48 (White et al. 2022) *Science* 2022 Aug 19;377(6608):834-839.

RESPONSE: Thank you for finding this error, we have added the journal name to the aforementioned citation.

Reviewer #2 (Remarks to the Author):

I have reviewed this manuscript twice already and I think it has improved substantially. I do not have any further comments, except for requesting the authors and editors to ensure that the compiled data and code are publicly available in a permanent repository. While the paper itself is a valuable and interesting contribution, the dataset of growth records will be used in many other future studies if it is made available.

All the best and congrats on a great study.

Asta Audzijonyte

RESPONSE: Thank you for your constructive comments, which we believe have greatly improved our manuscript.

Reviewer #3 (Remarks to the Author):

Dear authors and Editor,

Thank you very much for the opportunity to review “Over a century of global decline in growth performance of commercially valuable fishes” for a second time. Seeing the responses to my own comments, and those of other reviewers (especially R#1 whose comments I was asked to look at specifically), I still think it’s a great manuscript building on a massive data collection effort that presents a much needed global-scale empirical assessment of trends in growth/size.

RESPONSE: Thank you for your careful attention to our work.

Here are my comments. Apologies if they are scattered, I’m replying both to your replies, R#1 replies, and new things in the text!

1. Line 352: I appreciate the effort to translate growth performance into something more relatable, and I fully understand it is inherently challenging given the same change can be due to a change in K , L_{inf} , or both. You chose to hold one constant to solve for the remaining. I wonder if you could add a figure of how those changes in growth curves would look like. If L_{inf} or K matters for the growth trajectory, so a change in growth performance alone doesn’t say much about which growth trajectories become more common under fishing (as an example). One thing they have in common though is that for a given age, the size is reduced. Perhaps you could add a figure if you have room for that, like the one below, to help show that while there are differences (depending on if L_{inf} or K flexible), there are commonalities in the growth trajectory. See the figure below (code at the end of the document).

RESPONSE: Thank you for the great suggestion and associated R code. We have now added an additional figure to the supplemental materials (now denoted as Fig. S2) and the following text:

Notably, in both cases (i.e. a change in L_{∞} or a change in K), the size for a given age is reduced (Fig. S2).

This is my only follow up on the points I raised in my first review. I buy all the other changes and appreciate the effort but into this major review.

2. Reply to R#1 comment 1: Just a note; in A) I believe that the reviewer mean size-truncation at the population level, not changes in species composition. That said, it's not immediately clear to me how fishing removing large and old individuals affects growth parameters if there's no genetic effect (mechanism B). I don't think I've seen a simulation study on that; how sampling young fish leads to biased growth parameters. The Rosa Lee effect could lead to that, but I that's what the reviewer had in mind because it's not about truncation per se. And you already discuss the Rosa Lee effect, which I think is the most relevant here given your findings. It also doesn't invoke criticised mechanisms or genotypic change.

RESPONSE: Thank you for clarifying the comments provided by R1. We agree that the two aforementioned mechanisms that can affect a population's growth performance – size truncations via size-selective fisheries (including the Rosa Lee phenomenon) and subsequent selective pressures – are not mutually exclusive, with the latter a likely consequence of the former. Size-selective fisheries that remove the largest and oldest individuals from a stock can affect growth parameters in two ways: (1) the remaining population exhibits smaller body sizes, which can bias estimated growth performance values (Figure S1) and/or (2) fast-growing young individuals recruit into the fishery earlier and are selectively removed alongside older fishes, leaving a residual population disproportionately composed of slow-growing individuals (Rosa Lee phenomenon). These patterns can then be reinforced by genetic effects arising from natural selection, which could manifest as long-term declines in growth performance (i.e. over multiple generations). Note that we have discussed these mechanisms as plausible in the previously revised manuscript, but have also modified one of the sentences in this section to clarify these points. It now reads as:

Therefore, the declines in growth performance of commercially valuable fishes over time could represent the interplay between two synergistic processes. First, an unfished, residual stock is remaining with smaller individuals from size-selective fishing, which may be dominated by slow-growing individuals (via the “Rosa Lee” phenomenon⁴³). Second, these demographic changes are then further reinforced and amplified by fisheries-induced selection against larger body sizes, which would manifest as lower growth performance values in subsequent generations^{45–48}.

3. Line 29: I would refrain from calling K a growth rate. Rather say von Bertalanffy growth coefficient (K) at the first mention, and then just growth coefficient. This because it has unit per time (not size per time).

RESPONSE: Done.

4. Line 93: I don't really follow the sentence starting with “Therefore”, perhaps check the grammar!

RESPONSE: We have modified this sentence. It now reads as:

This global decline is most likely driven by fishes that are growing to relatively smaller sizes and/or at slower rates (i.e. intraspecific changes), and less likely driven by compositional shifts in the species sampled (i.e. interspecific changes; Fig. S3).

5. Line 133: This is true! I'd argue also that even managed fisheries without overfishing (but maximally fished) still have exploitation rates that would give rise to fishing induced evolution and/or Rosa Lee effects (just to strengthen your argument; it's not overfishing per se, but the magnitude of prolonged fishing that would lead to demographic changes).

RESPONSE: We have added an additional sentence to include the potential impact of prolonged fishing exposure. This sentence now reads as:

Additionally, some stocks that are effectively managed may not be overfished but may experience prolonged fishing pressures (i.e. maximally fished), which can similarly lead to fishing-induced demographic changes²⁰.

Reviewer #4 (Remarks to the Author):

I found, in the revised MS, my comments have been fully taken into account. Therefore, I recommend publication of this paper.

RESPONSE: Thank you for your assessment of our work.